# Peer review of "A Comprehensive Review of RFID and Bluetooth Security: Practical Analysis"

_technologies, doi:10.3390/technologies7010015_

Round 1

Reviewer 1 Report

The paper present some methods to exploit well-known vulnerabilites of RFID/NFC and Bluetooth.

In my opinion, the paper present some interesting considerations regarding the usage of open-source tools to demonstrate vulnerabilites. Although I can't identify the scientific and original contribution of the authors. I expected to see at least some proposals on fixing the identified problems.

Author Response

COVER LETTER

Authors: Santiago Figueroa Lorenzo ([email protected]), Javier Añorga Benito ([email protected]), Pablo García Cardarelli ([email protected]), Jon Alberdi Garaia ([email protected]) and Saioa Arrizabalaga Juaristi ([email protected]).

Research Center: University of Navarra, CEIT-IK4

Address: Parque Tecnológico de San Sebastián, Paseo Mikeletegi, Nº 48 20009, Donostia - San Sebastián

Manuscript: technologies-412438

New Title: “A comprehensive review of RFID and Bluetooth security: practical analysis”

Dear Reviewer 1:

We are pleased to resubmit an original research article entitled “A comprehensive review of RFID and Bluetooth security: practical analysis” by Santiago Figueroa Lorenzo, Javier Añorga Benito, Pablo García Cardarelli, Jon Alberdi Garaia and Saioa Arrizabalaga Juaristi for consideration for publication in the Journal Technologies (ISSN 2227-7080).

Important changes have been made for this resubmission which will be listed in the order in which they have been added. Finally, the answers given to the reviewer's comments will be analyzed:

·        The title has been changed to one that we consider best reflects the scope of the article.

·        The introduction has been modified by adding references and specifying the organization of the article where for each section is analyzed first the RFID technology, then NFC and finally Bluetooth.

·        The number of sections has changed. Two sections have been added, one dedicated to carrying out a review of vulnerabilities, threats and risks for each technology and another dedicated to the countermeasures to be applied.

·        As mentioned section 3 has been added. A detailed review of the main vulnerabilities, threats and risks are listed for each of the technologies. In order to develop this work, many bibliographic references were consulted, as well as platforms such as NIST, CVE-Mitre and CVE-Details. In addition, a column has been added to indicate to which of the vulnerabilities a practical analysis will be performed.

·        In section 4 two new open source tools have been added: Btlejuice and Bleah. Both for Bluetooth analysis.

·        Section 5 presents 2 new added architectures, one for BtleJuice to deploy the MITM and one for Bleah to deploy the fuzzing attack.

·        The practical analysis in section 6 has been related to the vulnerabilities presented in section 3. In this way, the tables shown below allow a very concise understanding of the new analyses carried out. Significantly, three new vulnerabilities were exploited for Bluetooth.

·           

Security risk and threat

RFID Tag

NFC Tag

Spoofing

HID ProxCard II

Cloning Tags

EM4100

Emulate cards

T5577

Eavesdropping

Mifare Classic

Password Decoding or Crypto attacks

Mifare Classic

Table 2. Practical analysis at RFID and NFC tags

Vulnerability No.

Security Test

Device

1

Security mode 1 Level 1, no security

CC2650STK

2

Just Works, MITM attack

CC2650STK

3

Just Works, Eavesdropping attack

CC2650STK

8

Passkey, Eavesdropping attack

CC2540DK

13

Security Services limited, Fuzzing attack

CC2650STK

Table 3. Practical analysis at Bluetooth devices

·        Subsections 6.1.1, 6.1.2 and 6.1.5 have been added with content about others Bluetooth vulnerabilities exploited as part of the practical analysis.

·        Section 7 is added, including risk mitigation and counter measurements for each technology.

·        The conclusions have been detailed mentions the contributions that the article present. In addition, the conclusions have been by adding new lines of research that are being worked on such as the case of the security analysis in NFC DESfire EV1 and EV2 cards, the security analysis for RFID tags ALN-9640 Squiggle and Monza R6 - P, as well as the new security tests that are being carried out for the security mode LE Secure and Bluetooth mesh.

·        A total of 88 references have been consulted in the manuscript. The references have been modified with all fields completes.

Responses to Reviewer #1

Thank you for the review and your comments. We consider that thanks to the comments, this manuscript has improved in content and value for the reader. We hope that the new information, methodologies and analysis will meet your expectations. In addition, was included a section with risk mitigation and counter measurements.

Reviewer 2 Report

The paper analyses security issues of RFID and Bluetooth technologies, by using two different tools  (Ubertooth and PM3).

The paper lacks in novelty since hundreds of papers and books can be found dealing with security with these two technologies. The authors themselves have already presented the results about RFID in a previous paper (some of the figures present in this paper have in fact already been used in this  one):

S. Figueroa, J. F. Carías, J. Añorga, S. Arrizabalaga and J. Hernantes, "A RFID-based IoT Cybersecurity Lab in Telecommunications Engineering," 2018 XIII Technologies Applied to Electronics Teaching Conference (TAEE), La Laguna, 2018, pp. 1-8.

While the paper aims at presenting the tests tools, they are based on already existing tools and the actual novelty of their proposal is not clear.

Moreover the paper is not well structured. The description about RFID and Bluetooth, two well known technologies, is overlong, while no state of the art is presented. The most part of the references deal with the technologies employed rather than with similar works.

I believe that the paper is not suitable for publication since no actual novelty is introduced.

Author Response

COVER LETTER

Authors: Santiago Figueroa Lorenzo ([email protected]), Javier Añorga Benito ([email protected]), Pablo García Cardarelli ([email protected]), Jon Alberdi Garaia ([email protected]) and Saioa Arrizabalaga Juaristi ([email protected]).

Research Center: University of Navarra, CEIT-IK4

Address: Parque Tecnológico de San Sebastián, Paseo Mikeletegi, Nº 48 20009, Donostia - San Sebastián

Manuscript: technologies-412438

New Title: “A comprehensive review of RFID and Bluetooth security: practical analysis”

Dear Reviewer 2:

We are pleased to resubmit an original research article entitled “A comprehensive review of RFID and Bluetooth security: practical analysis” by Santiago Figueroa Lorenzo, Javier Añorga Benito, Pablo García Cardarelli, Jon Alberdi Garaia and Saioa Arrizabalaga Juaristi for consideration for publication in the Journal Technologies (ISSN 2227-7080).

Important changes have been made for this resubmission which will be listed in the order in which they have been added. Finally, the answers given to the reviewer's comments will be analyzed:

·        The title has been changed to one that we consider best reflects the scope of the article.

·        The introduction has been modified by adding references and specifying the organization of the article where for each section is analyzed first the RFID technology, then NFC and finally Bluetooth.

·        The number of sections has changed. Two sections have been added, one dedicated to carrying out a review of vulnerabilities, threats and risks for each technology and another dedicated to the countermeasures to be applied.

·        As mentioned section 3 has been added. A detailed review of the main vulnerabilities, threats and risks are listed for each of the technologies. In order to develop this work, many bibliographic references were consulted, as well as platforms such as NIST, CVE-Mitre and CVE-Details. In addition, a column has been added to indicate to which of the vulnerabilities a practical analysis will be performed.

·        In section 4 two new open source tools have been added: Btlejuice and Bleah. Both for Bluetooth analysis.

·        Section 5 presents 2 new added architectures, one for BtleJuice to deploy the MITM and one for Bleah to deploy the fuzzing attack.

·        The practical analysis in section 6 has been related to the vulnerabilities presented in section 3. In this way, the tables shown below allow a very concise understanding of the new analyses carried out. Significantly, three new vulnerabilities were exploited for Bluetooth.

·           

Security risk and threat

RFID Tag

NFC Tag

Spoofing

HID ProxCard II

Cloning Tags

EM4100

Emulate cards

T5577

Eavesdropping

Mifare Classic

Password Decoding or Crypto attacks

Mifare Classic

Table 2. Practical analysis at RFID and NFC tags

Vulnerability No.

Security Test

Device

1

Security mode 1 Level 1, no security

CC2650STK

2

Just Works, MITM attack

CC2650STK

3

Just Works, Eavesdropping attack

CC2650STK

8

Passkey, Eavesdropping attack

CC2540DK

13

Security Services limited, Fuzzing attack

CC2650STK

Table 3. Practical analysis at Bluetooth devices

·        Subsections 6.1.1, 6.1.2 and 6.1.5 have been added with content about others Bluetooth vulnerabilities exploited as part of the practical analysis.

·        Section 7 is added, including risk mitigation and counter measurements for each technology.

·        The conclusions have been detailed mentions the contributions that the article present. In addition, the conclusions have been by adding new lines of research that are being worked on such as the case of the security analysis in NFC DESfire EV1 and EV2 cards, the security analysis for RFID tags ALN-9640 Squiggle and Monza R6 - P, as well as the new security tests that are being carried out for the security mode LE Secure and Bluetooth mesh.

·        A total of 88 references have been consulted in the manuscript. The references have been modified with all fields completes.

Responses to Reviewer #2

1.      Thank you for the review and your comments. We consider that thanks to the comments, this manuscript has improved in content and value for the reader. We hope that the new information, methodologies and analysis will meet your expectations.

2.      We consider that once the focus has changed through the new title, the detailed review of vulnerabilities, risks and threats, using updated bibliography and platforms recognized in the world of cybersecurity, associating the practical analysis to some of the vulnerabilities mentioned and adding countermeasures and risk mitigation can giving the novelty to the article.

3.      The new approach of the article as a review, allows to detail the practical analysis from 6 architectures and 10 vulnerabilities analyzed using tools not only open hardware, but also open software such as BtleJuice for the MITM attack and Bleah for the fuzzing attack.

We also believe that the approach of countermeasures and risk mitigation is very important since elements have been established to prevent the vulnerabilities and threats mentioned.

Reviewer 3 Report

The authors of this paper evaluate the security of radio frequency identification (RFID) and Bluetooth devices. Thus, using open source and open hardware tools, particularly ProxMark3 and Ubertooth, security tests are carried out when recovering encryption keys. Thus, by analyzing the security of these devices considering the topology, architecture and the exchange of messages, the authors provide a very interesting and practical study of how the security of the devices analyzed can be compromised.

The devices used for the analysis in this paper represent a very good choice for both types of technologies. However, a more detailed description of how this could be generalized to alternative devices of the same technology would be highly desirable.

The paper is well structured. Nevertheless, it must be fully revised for English grammar. Only the Introduction and Conclusions sections are well written, but the rest of the paper really needs to be improved.

Author Response

COVER LETTER

Authors: Santiago Figueroa Lorenzo ([email protected]), Javier Añorga Benito ([email protected]), Pablo García Cardarelli ([email protected]), Jon Alberdi Garaia ([email protected]) and Saioa Arrizabalaga Juaristi ([email protected]).

Research Center: University of Navarra, CEIT-IK4

Address: Parque Tecnológico de San Sebastián, Paseo Mikeletegi, Nº 48 20009, Donostia - San Sebastián

Manuscript: technologies-412438

New Title: “A comprehensive review of RFID and Bluetooth security: practical analysis”

Dear Reviewer 3:

We are pleased to resubmit an original research article entitled “A comprehensive review of RFID and Bluetooth security: practical analysis” by Santiago Figueroa Lorenzo, Javier Añorga Benito, Pablo García Cardarelli, Jon Alberdi Garaia and Saioa Arrizabalaga Juaristi for consideration for publication in the Journal Technologies (ISSN 2227-7080).

Important changes have been made for this resubmission which will be listed in the order in which they have been added. Finally, the answers given to the reviewer's comments will be analyzed:

·        The title has been changed to one that we consider best reflects the scope of the article.

·        The introduction has been modified by adding references and specifying the organization of the article where for each section is analyzed first the RFID technology, then NFC and finally Bluetooth.

·        The number of sections has changed. Two sections have been added, one dedicated to carrying out a review of vulnerabilities, threats and risks for each technology and another dedicated to the countermeasures to be applied.

·        As mentioned section 3 has been added. A detailed review of the main vulnerabilities, threats and risks are listed for each of the technologies. In order to develop this work, many bibliographic references were consulted, as well as platforms such as NIST, CVE-Mitre and CVE-Details. In addition, a column has been added to indicate to which of the vulnerabilities a practical analysis will be performed.

·        In section 4 two new open source tools have been added: Btlejuice and Bleah. Both for Bluetooth analysis.

·        Section 5 presents 2 new added architectures, one for BtleJuice to deploy the MITM and one for Bleah to deploy the fuzzing attack.

·        The practical analysis in section 6 has been related to the vulnerabilities presented in section 3. In this way, the tables shown below allow a very concise understanding of the new analyses carried out. Significantly, three new vulnerabilities were exploited for Bluetooth.

·           

Security risk and threat

RFID Tag

NFC Tag

Spoofing

HID ProxCard II

Cloning Tags

EM4100

Emulate cards

T5577

Eavesdropping

Mifare Classic

Password Decoding or Crypto attacks

Mifare Classic

Table 2. Practical analysis at RFID and NFC tags

Vulnerability No.

Security Test

Device

1

Security mode 1 Level 1, no security

CC2650STK

2

Just Works, MITM attack

CC2650STK

3

Just Works, Eavesdropping attack

CC2650STK

8

Passkey, Eavesdropping attack

CC2540DK

13

Security Services limited, Fuzzing attack

CC2650STK

Table 3. Practical analysis at Bluetooth devices

·        Subsections 6.1.1, 6.1.2 and 6.1.5 have been added with content about others Bluetooth vulnerabilities exploited as part of the practical analysis.

·        Section 7 is added, including risk mitigation and counter measurements for each technology.

·        The conclusions have been detailed mentions the contributions that the article present. In addition, the conclusions have been by adding new lines of research that are being worked on such as the case of the security analysis in NFC DESfire EV1 and EV2 cards, the security analysis for RFID tags ALN-9640 Squiggle and Monza R6 - P, as well as the new security tests that are being carried out for the security mode LE Secure and Bluetooth mesh.

·        A total of 88 references have been consulted in the manuscript. The references have been modified with all fields completes.

Responses to Reviewer #3

Thank you for the review and your comments. We consider that thanks to the comments, this manuscript has improved in content and value for the reader. We hope that the new information, methodologies and analysis will meet your expectations.

The new approach of the article as a review, allows to detail the practical analysis from 6 architectures and 10 vulnerabilities analyzed using tools not only open hardware, but also open software such as BtleJuice for the MITM attack and Bleah for the fuzzing attack. All this with the aim of providing not only open hardware tools, but also alternative open software tools.

In addition, the grammar of the entire manuscript was revised, considering that it has improved.

Round 2

Reviewer 1 Report

First I would like to thank the authors for taking into consideration my observations.

In my opinion the present form of the paper fullfils the requirements for publishing in a scientfic journal. The authors present the vulnerabilites of RFID and Bluetooth technologies in a comprehensive bibliography study, which is supported by an experimental work. The experimental part is easily to be reporduced since commercial and open-source tools were used.

The conclusion is supported by the article contents.

The bibliography is relevant and up to date.

Author Response

Dear Reviewer 1:

We are pleased to resubmit an original research article entitled “A comprehensive review of RFID and Bluetooth security: practical analysis” by Santiago Figueroa Lorenzo, Javier Añorga Benito, Pablo García Cardarelli, Jon Alberdi Garaia and Saioa Arrizabalaga Juaristi for consideration for publication in the Journal Technologies (ISSN 2227-7080).

Some changes have been made for this resubmission. Finally, the answers given to the reviewer's comments will be analyzed:

1.      English grammar has been revised.

2.      Repetitive phrases and words have been deleted or changed.

3.      Details with acronyms have been fixed.

4.      Some figure’s captions have been fixed.

5.      Some table’s number have been fixed.

6.      One command’s number has been fixed.

7.      Figures have been enlarged.

Responses to Reviewer #1

1.      Thank you for the review and your comments.

Reviewer 2 Report

The paper has been notably improved from its previous version. Most important, its current review form is more suitable to the presented topic.

The content of the paper is sound and interesting. Nevertheless, some corrections are required prior to its publication:

The paper needs a revision for what concerns the English: I suggest a proofread by an English native speaker. Moreover, several typos are found throughout the text (e.g. line 19: affecting to both -> affecting both, line 87: compatibility -> compatible, line 275: RPBi -> RBPi, etc...)

In several sections you write sentences like "In order to install and getting started...": since you are writing a scientific paper and not a tutorial you should avoid this kind of sentences and just add the reference to the manual

Some Figures are too small, in particular Fig 2 e Fig. 3., and should be enlarged

In line 212 you write "two open source and open hardware tools" but you list four of them.

Author Response

Dear Reviewer 2:

We are pleased to resubmit an original research article entitled “A comprehensive review of RFID and Bluetooth security: practical analysis” by Santiago Figueroa Lorenzo, Javier Añorga Benito, Pablo García Cardarelli, Jon Alberdi Garaia and Saioa Arrizabalaga Juaristi for consideration for publication in the Journal Technologies (ISSN 2227-7080).

Some changes have been made for this resubmission. Finally, the answers given to the reviewer's comments will be analyzed:

1.      English grammar has been revised.

2.      Repetitive phrases and words have been deleted or changed.

3.      Details with acronyms have been fixed.

4.      Some figure’s captions have been fixed.

5.      Some table’s number have been fixed.

6.      One command’s number has been fixed.

7.      Figures have been enlarged.

Responses to Reviewer #2

1.      Thank you for the review and your comments. We consider that thanks to the comments, this manuscript has improved in content and value for the reader. We hope that the changes realized will meet your expectations.

2.      The paper needs a revision for what concerns the English: I suggest a proofread by an English native speaker. Moreover, several typos are found throughout the text (e.g. line 19: affecting to both -> affecting both, line 87: compatibility -> compatible, line 275: RPBi -> RBPi, etc...).

Many changes were made. The mistakes mentioned, have been specifically corrected at:

·        Line 19: Phrase modified to “affecting both”.

·        Line 85: Word changed: “compatible”.

·        Line 326: Acronym changed: “RBPi”

3.      In several sections you write sentences like "In order to install and getting started...": since you are writing a scientific paper and not a tutorial you should avoid this kind of sentences and just add the reference to the manual.

·        Line 234: Phrase modified: “getting started”.

·        Line 245: Phrase modified: “getting started”.

·        Line 261: Phrase modified: “getting started”.

4.      Some Figures are too small, in particular Fig 2 e Fig. 3., and should be enlarged

·        Figures have been enlarged.

5.      In line 212 you write "two open source and open hardware tools" but you list four of them.

                          ·       Line 208: Word changed: “four”.

This manuscript is a resubmission of an earlier submission. The following is a list of the peer review reports and author responses from that submission.

Round 1

Reviewer 1 Report

In this work, the authors tested the security challenges of two broadly used technologies, RFID and Bluetooth, are analyzed from a practical point of view. The architecture, the methodology and the HW/SW which have been used for the analysis are nicely detailed and explained.

 Results show that information confidentiality is exploited in both technologies by using open source and open hardware tools (Proxmark3 and Ubertooth).

·       I would like to have a good analysis for the reason why the confidentiality is broken and what are the suggested solutions to fix this issue in terms of architecture.

·       The conclusion is simple. Please have more details.

·       Is there a plan for future work?

Author Response

We appreciate all his comments and we thank him all for his precious time. The document uploaded tries to answer each of the suggestions and comments made by the reviewer.

Reviewer 2 Report

The paper focuses on presenting the main security vulnerabilities of RFID/NFC and Bluetooth technologies.  The first sections of the paper are dedicated to describing in detail the two technologies, that have applications in IoT.

The obtained results sustain the conclusion that RFID and Bluetooth are vulnerable in regard to confidentiality if the data is intercepted by an attacker with adequated tools.

The bibliography is updated and relevant for the topic of the paper.

I noticed that most of the figures are numbered as Figure N a), but each figure has only one element. I suggest the authors remove the letter a) from figures legend and for paper text.

In the conclusions, in line 270 please replace "y" with "and".

The English language should be revised. 

Author Response

(The authors gave the same response as above.)

Reviewer 3 Report

This article summarizes two well-known weaknesses (one in MIFARE Classic and one in BLE pairing) and how to use the tools to validate those weaknesses.

There is no novelty in this article. Instead, the authors simply repeat (and validate) the weaknesses discovered by Nohl, Garcia, de Koning Gans, Verdult et al. in the MIFARE Classic cipher and PRNG. These weaknesses are well-known for more than 10 years now. Btw. I'm not sure why this would relate to other NFC technologies (particularly to NFC tags that store freely readable information and typically don't use any communication encryption at all).

The authors do the same for Bluetooth, where they describe and reproduce a well-known weakness in BLE pairing process. This weakness including the crackle tool that the authors use and present was found and first presented by Mike Ryan at WOOT in 2013. Again, there's nothing novel about this.

Finally, the authors do not draw any new conclusions from their tests of the existing tools.

Hence, I suggest this article to be rejected. I don't see any point for even a major revision. The article is not a research paper, it might be suitable for a computer magazine though.

Author Response

(The authors gave the same response as above.)

Round 2

Reviewer 1 Report

Authors have addressed all my questions!

Reviewer 3 Report

The new revision does not significantly differ from the previous version. Hence, my previous comments regarding lack of novelty and scientific relevance (problems discovered and partially solved years ago). are still valid.

In their cover letter, the authors argued that their contribution is bringing the necessary tools into one platform (Raspberry Pi). Since all the used tools are open-source and readily available for that architecture, this still does not seem like a novel contribution to me. Moreover, the authors state that "as the baseline, methodology and steps have been described in detail, so that other novel researchers can also use and reproduce this analysis in their initial steps. We think this is also another important contribution." However, with regard to the described attacks, this is exactly what the original authors of those attacks already did in their papers. So there is no novel contribution about this either. Finally, the paper is too selective (only two specific attacks were chosen) to be considered as a literature review paper.

Therefore, I still suggest the paper to be rejected.

A few more things:

- There is no NFC-C. Did you mean NFC-F?

- While MIFARE Classic was the starting point for NFC (actually the whole MIFARE family was), MIFARE Classic itself is not NFC since it does not follow the framing, etc. defined by the NFC Forum tag specifications. It may be considered NFC-compatible though.

- In the conclusion the authors claim that they intend to perform similar analysis for other NFC cards. However, 3 out of the 4 mentioned card technologies are not even NFC. EM4100, T5577, and HID Prox even operate on a different carrier frequency than NFC (125kHz instead of 13.56MHz). Hence, they are not even NFC-compatible!

- The list of references is poorly formatted. E.g. for many entries, names of corporations were abbreviated with initials (this should only happen for person names!). Several author names are misspelled. Etc.